# RrED: Black-box Unsupervised Domain Adaptation via Rectifying-reasoning Errors of Diffusion

**Yuwu Lu**[*][†]
School of Artificial Intelligence
South China Normal University
Foshan, Guangdong, China
luyuwu2008@163.com

**Chunzhi Liu**[†]
School of Artificial Intelligence
South China Normal University
Foshan, Guangdong, China
2023024323@m.scnu.edu.cn

## Abstract

Black-box Unsupervised Domain Adaptation (BUDA) aims to transfer source domain knowledge to an unlabeled target domain, without accessing the source data or trained source model. Recent diffusion models have significantly advanced the ability to generate images from texts. While they can produce realistic visuals across diverse prompts and demonstrate impressive compositional generalization, these diffusion-based domain adaptation methods focus solely on composition, overlooking their sensitivity to textual nuances. In this work, we propose a novel diffusion-based method, called *Rectifying-reasoning Errors of Diffusion* (RrED) for BUDA. RrED is a two-stage learning strategy under diffusion supervision to effectively enhance the target model via the decomposed text and visual encoders from the diffusion model. Specifically, RrED consists of two stages: *Diffusion-Target model Rectification* (DTR) and *Self-rectifying Reasoning Model* (SRM). In DTR, we decouple the image and text encoders within the diffusion model: the visual encoder integrates our proposed feature-sensitive module to generate inferentially-enhanced visuals, while the text encoder enables multi-modal joint fine-tuning. In SRM, we prioritize the BUDA task itself, leveraging the target model's differential reasoning capability to rectify errors during learning. Extensive experiments confirm that RrED significantly outperforms other methods on four benchmark datasets, demonstrating its effectiveness in enhancing reasoning and generalization abilities.

## 1 Introduction

To address domain shift in training deep neural networks [1], domain generalization (DG) methods [2, 3] only use source data for model learning to achieve generalization. However, in scenarios with accessible target samples, domain adaptation (DA) methods [4–15] show a significant performance advantage. Traditional unsupervised domain adaptation (UDA) methods [4–8] focus on adapting models trained on a fully labeled source domain to an unlabeled target domain, aiming to alleviate the constraints of data collection and annotation. However, in scenarios like personal medical records, privacy-preserving policies restrict access to source data, thus limiting the application of UDA techniques. To address this, source-free domain adaptation (SFDA) methods [9–12] have been recently introduced, assuming that only unlabeled target domain data and a pre-trained source model are available during the adaptation process. Even though SFDA methods lower the possibility of privacy leaks by utilizing the pre-trained source model rather than source data, [15] found that certain generation techniques like [13, 14] have the potential to reconstruct the source data through

---

[*]*Corresponding author.*
[†]*Both authors contributed equally to this work.*

39th Conference on Neural Information Processing Systems (NeurIPS 2025).

learning from the source model. In comparison to other UDA settings, black-box unsupervised domain adaptation (BUDA) offers enhanced data privacy protection along with greater flexibility in portability. BUDA adapts a model by leveraging the unlabeled target data and a black-box predictor trained on the source domain, *e.g.*, an API service in the cloud [15], to avoid privacy and safety problems caused by data and model leakage.

Recent mainstream BUDA methods [15–18] follow a self-distillation process: distilling source knowledge and fine-tuning the model for the target domain. This process relies on high-reliability samples to suppress the negative impact of low-reliability ones. However, AEM [19] observes that distillation-based methods selectively ignore those samples that are classified as low reliability, resulting in underlying structural information from low-reliability samples not being utilized. To alleviate this problem, AEM introduces the multi-modal model CLIP [20] as an external prompt to extract semantic knowledge. However, AEM primarily forces the target model to align with CLIP, overlooking the further exploration of the target model. Similar to CLIP, diffusion models [21–23] also use multi-modal techniques, which represent a new category of likelihood-based generative models that introduce iterative noise and denoising processes to model the

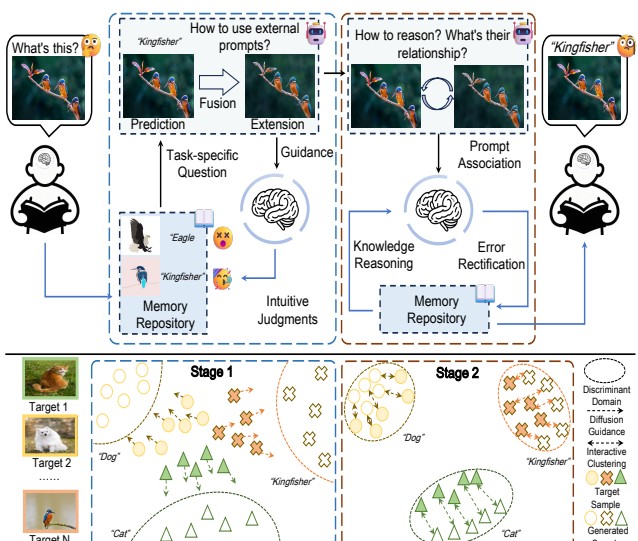

Figure 1: Conceptual figure of our RrED. Above: our RrED simulates the human decision-making process under the guidance of external knowledge. Below: in stage 1, RrED aligns the decision boundaries of the diffusion model under guidance; in stage 2, RrED enhances the model's self-reasoning ability by rectifying the errors among different versions.

data distribution. Compared to other multi-modal models, diffusion models not only have rich semantic knowledge but also can stably generate diverse images. Previous diffusion-based DA methods [24–26] focus on enhancing sample distribution generalization via the image encoder, with limited exploration of the text encoder, and are restricted in scenarios where source samples are inaccessible due to protective policies. In the BUDA setting, such stable generation helps enhance model generalization by providing consistent image-level augmentation. How to effectively leverage diffusion models in BUDA tasks to guide the target model in enhancing its reasoning ability while preventing its potential negative effects? This is the key problem that needs to be addressed in this research.

In the field of neuroscience [27, 28], the human decision-making process is typically regarded as an interaction between two stages: Stage 1 unconsciously generates intuitive responses but tends to exhibit cognitive biases and struggles with complex tasks like mathematical reasoning or weighing pros and cons; Stage 2 relies on domain knowledge for deliberate reasoning, handling complex problems more accurately but at a slower pace. Recent works [29, 30] have observed that discrepancies between multi-stage decision-making can introduce potential errors in reasoning. To address this issue, these works introduce a knowledge base to guide the intuitive learning process during the intuitive response process, leveraging domain knowledge to identify and correct potential errors in the neural network's output during the reasoning phase, thereby producing outputs consistent with the knowledge base.

Inspired by these works, we propose a novel diffusion-based method, named *Rectifying-reasoning Errors of Diffusion* (RrED), which is the first work that applies the diffusion model to high-security BUDA tasks innovatively. Specifically, RrED addresses the limitations of previous diffusion-based methods by continually fine-tuning the text encoder while learning from the diffusion model. RrED is composed of two stages: *Diffusion-Target model Rectification* (DTR) that performs separation learning of the diffusion image encoder and text encoder and task-specific fine-tuning of the text encoder, *Self-rectifying Reasoning Model* (SRM) that leverages the differential reasoning ability of the

Table 1: Comparison of different settings. Diffusion-based DA relies on both labeled source and unlabeled target data, guided by an external diffusion model. Black-box DA only relies on the unlabeled target data and the predicted labels from a black-box predictor, thus offering better data privacy at the cost of partial performance. Our RrED follows Black-box DA setting for training with a diffusion model incorporated, achieving performance improvement while maintaining high-level data privacy protection.

| Setting | Source data | Source model | Predicted target labels | Target data | External prompt | Privacy risk |
|---|---|---|---|---|---|---|
| DG | ✓ | ✓ | × | × | × | Medium |
| Traditional DA | ✓ | ✓ | ✓ | ✓ | × | High |
| Source-free DA | × | ✓ | ✓ | ✓ | × | Medium |
| Black-box DA | × | × | ✓ | ✓ | × | Low |
| Diffusion-based DA | ✓ | ✓ | ✓ | ✓ | ✓ | High |
| Our RrED | × | × | ✓ | ✓ | ✓ | Low |

target model and samples generated by the fine-tuned diffusion model to correct errors in the learning process. RrED introduces the diffusion model as a knowledge base to rectify the memory repository of BUDA. As shown in Figure 1, in DTR, the target model (modeled as the human brain) receives the guidance information from the diffusion model to make intuitive judgments about the task-specific information of BUDA and feeds back the discrepancies between predictions to fine-tune the diffusion model. In SRM, after being guided by the diffusion model, the target model conducts thoughtful comparative reasoning and error correction on the task-specific information. The purpose of the two stages is to fine-tune the text encoder of the diffusion model and use the fine-tuned diffusion model to further improve the discriminative ability of the target model. Experimental results demonstrate that RrED significantly outperforms the previous SOTA methods on four benchmarks, confirming its effectiveness in enhancing the model's reasoning and generalization abilities.

Our contributions are summarized as follows:

• We observe some weaknesses in existing DA methods and address them by proposing a novel method, named RrED, which introduces the diffusion model into the BUDA setup and strengthens the target model's reasoning ability through our two-stage learning.

• Inspired by the improved human decision-making process, RrED is designed to consist of two stages, namely DTR and SRM. DTR guides the target model's learning process by rectifying diffusion model reasoning errors and leveraging its knowledge. SRM corrects errors in the learning process by leveraging the differential reasoning ability of the target model and samples generated by the fine-tuned diffusion model.

• To evaluate the effectiveness of RrED, we conduct extensive experiments, achieving SOTA performance on four benchmarks. Ablation studies further highlight the contributions of each component and provide a detailed analysis of the relationship among them.

## 2 Related Works

**Domain Adaptation.** The challenge of unsupervised domain adaptation (UDA) resides in transferring the knowledge from the labeled source domain to a related yet distinct unlabeled target domain. Recently, UDA has been the subject of widespread research in diverse deep learning tasks, including image classification [4, 5, 31], semantic segmentation [7, 32, 33], object detection [6, 34, 35], and time series forecasting [8, 36, 37]. However, UDA relies on access to both the labeled source domain and the unlabeled target domain during training, which becomes restrictive under privacy-preserving policies that limit source data availability. To overcome this, source-free domain adaptation (SFDA) [9, 38, 39, 10] enhances privacy protection by requiring only the trained source model and unlabeled target data. Although SFDA methods mitigate privacy data breaches to some extent, recent works [15, 40] highlight the risks of exposing the training white-box model in SFDA, as reverse generation techniques [13, 14] can exploit this vulnerability.

**Black-box Unsupervised Domain Adaptation.** BUDA has no need to access the source data or trained model, which enhances data privacy protection more effectively than other DA settings, reducing the risk of data breaches. Early work LNL-KL [41] proposes a noisy label learning approach using soft labels. Recent work DINE [15] first distills knowledge to encourage source-target

class alignment and then fine-tunes the distilled model to match the target distribution, using the reliable knowledge from distillation to cluster unreliable samples during fine-tuning. Building on the self-distillation process, recent methods [16, 17] partition the target domain into high- and low-reliability subdomains, and align their distribution discrepancies. BiMem [40] performs information discrimination between useful and irrelevant information, emphasizing prioritized learning of useful samples while roughly aggregating irrelevant ones. RFC [18] further introduces neighborhood clustering into [15, 16] to avoid minority class forgetting. Moreover, AEM [19] first introduces a multi-modal model CLIP [20] as an external prompt into BUDA, utilizing CLIP's rich semantic knowledge to conduct feature alignment of the target domain model.

**Diffusion Models in Domain Adaptation.** Diffusion models [22, 21, 42] use a parameterized Markov chain to transform noise from a common distribution to a target distribution. Recently, diffusion models have been applied across various tasks like image generation [26], video generation [23], and text-to-image generation [43], due to their support for the interaction and creation of text and image contents. In domain adaptation, some studies [25, 24, 26, 44] have recognized that diffusion can be used to improve the target model's generalization ability. DAD [25] learns additional source-style target samples by continuously synthesizing source and target domain images, gradually transforming the data distribution. SDA [44] maps source and target domain images to a synthesis space, transforming domain transfer into sample alignment in the synthesis space. However, we observe that current diffusion-based UDA methods rely on both source and target data, limiting their application when policy restrictions prevent access to source data or models. Moreover, these methods are based on image encoder synthesis and do not contribute to the development of text encoder in diffusion models. To solve these problems, RrED integrates diffusion into the BUDA task and guides the diffusion model's generation by fine-tuning the text encoder. As shown in Table 1, the respective processes and the differences among various settings are presented.

**The Definition of Reasoning Ability.** In human decision-making systems [27, 28], the definition of reasoning ability refers to the capacity of individuals to make further judgments about target objects by leveraging prior knowledge and logical analysis based on partial observations when confronted with complex information and dynamic environments. Similarly, in computer vision, the model's reasoning process mirrors human decision-making by utilizing existing knowledge and feature similarity computations to determine whether targets in complex scenes meet the task requirements. This perspective aligns with the explanations provided by Grad-CAM [45], where models reason about predicted image classes based on convolutional feature maps, analogous to how humans reason about image categories through attention maps. While the reasoning ability of Large Language Models (LLMs) typically refers to abstract and logical inference, in computer vision it focuses on identifying the input regions that most influence the model's decision to infer the most probable target class. The reasoning definition is based on the well-known Grad-CAM technique [45] in computer vision, and our work further integrates the observations of the human decision-making system to refine this definition and enhance the model's reasoning capabilities.

## 3 Proposed Method

We first define an unlabeled target domain $D_t = \{(x_i)\}_{i=1}^{N_t}$, where $N_t$ represents the number of unlabeled target domain data. In the BUDA setting, $D_t$ is uploaded to a black-box predictor (*i.e.*, cloud API service), which provides the hard predictions $P_s$ using a source model trained on the source domain $D_s$. $D_t$ and $D_s$ share an identical label distribution over $b$ classes, with a common label set $L = \{1, 2, \cdots, b\}$. Our goal is to enable the target model $\mathcal{M}_\theta$ to adapt in the target domain, parameterized by $\theta$ and composed of a feature extractor $f_\theta$ and a prediction classifier $c_\theta$. The feature extractor is defined as $f_\theta : x_i \rightarrow z_i \in \mathbb{R}^d$, where $d$ is the feature space dimension and $z_i$ is the $d$-dimensional transitional output. The prediction classifier is defined as $c_\theta : z_i \rightarrow y_i \in \mathbb{R}^b$, where $y_i$ is the prediction output of the target samples. The complete training process is shown in Figure 2.

### 3.1 Black-box Learning and Diffusion Process

**Task-specific Black-box Learning.** Before the training period, the black-box predictor exposes only an open API, allowing external clients to request predictions by uploading data. The source model and source samples remain inaccessible throughout the process, effectively preventing potential data leakage. Additionally, batching requests through a queue-based mechanism can improve response efficiency. Previous BUDA methods [15, 16, 40] stored the predicted labels of target samples returned

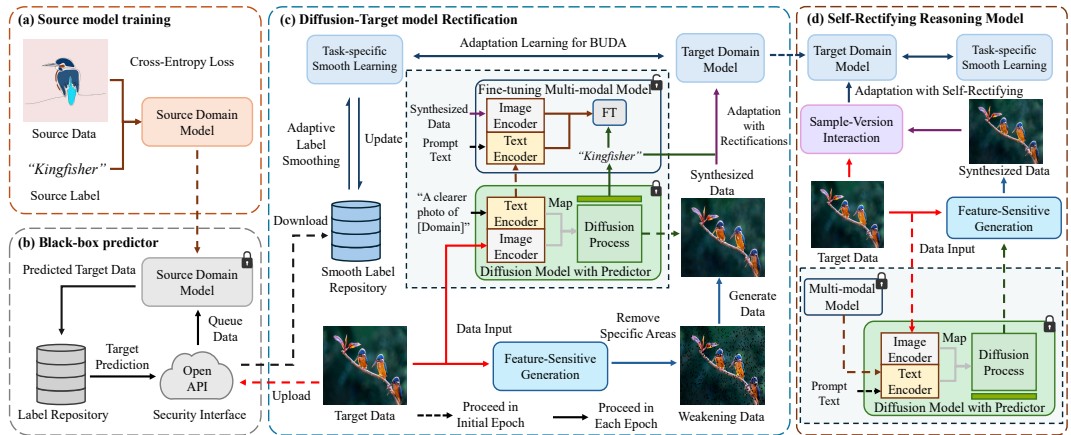

Figure 2: Overview of the whole training of RrED. According to the BUDA setting, (a) the source model is initially trained with standard procedures and transferred to a black-box predictor; (b) the black-box predictor then exposes a restricted API, allowing external clients to query only batches of hard target predictions through iterative requests. In our RrED, (c) DTR guides the target model's learning by correcting reasoning errors from the diffusion model and leveraging its semantic knowledge; (d) SRM corrects the reasoning error of the target model by leveraging the model's reasoning from predictive differences across versions.

by the open API into a smooth label repository and employed adaptive label smoothing (ALS) [15] to filter out some redundant and noisy information. The ALS updating $S(x_i)$ can be expressed as:

$$S(x_i) = \begin{cases} \frac{1}{N_t} \sum_{j=1}^{N_t} AdaLS(P_s^i), & beginning \\ \mu S(x_i) + (1-\mu)y_i, & otherwise \end{cases}, \quad (1)$$

where $AdaLS(P_s^i)$ is a function for initializing the smooth label repository in ALS [15]; $P_s^i$ represents the hard prediction of the $i$-th sample, obtained from the black-box predictor prior to training; and $\mu$ is set to 0.6 following [15, 16, 40], representing the static coefficient to stabilize the ALS updating.

During the learning process, the task-specific loss stably transfers knowledge from $S(x_i)$ to the target model to accomplish the BUDA task, ensuring efficient learning of target domain knowledge without forgetting source domain knowledge. The task-specific loss can be expressed as:

$$\mathcal{L}_{task} = -\min_{\mathcal{M}_\theta} \mathbb{E}_{x_i \in D_t}[D_{KL}(\mathcal{M}_\theta(x_i)||S(x_i))], \quad (2)$$

where $D_{KL}(\cdot)$ is the Kullback-Leibler divergence.

**Diffusion model for RrED.** In this work, we introduce an image encoder with fixed weights from CLIP [20] and leverage the predictor along with the image encoder during the diffusion-target model rectification to fine-tune the text encoder. Moreover, before the SRM period, the fine-tuned text encoder is fed back to the diffusion model to enable more controllable image generation and adapt the target model to the BUDA task. *The diffusion process and how the diffusion model is applied to RrED are described in detail in Appendix A.*

### 3.2 Diffusion-Target model Rectification

**Feature-Sensitive Generation (FSG).** During the first-stage training, the target model stably learns from the images generated by the diffusion model under the control of FSG. Despite the constraints imposed by the text encoder on the diffusion model, the image-generation process remains uncontrollable. To prevent the negative impacts of this uncontrollable factor, FSG needs to determine which regions should be composed of synthetic images, enhancing generalization without sacrificing discriminative ability. Specifically, FSG first evaluates the feature-sensitive regions of the model by evaluating the global image and comparing it with each local region, leveraging the local regions and their adjacent contextual information. The weight evaluation of each local area can be expressed as:

$$weight_{i,j,k} = max\{Tanh(\sum_{n=1}^b \sum_{l=1}^d \frac{z_{i,(j,k)}^l}{z_i} \cdot \frac{\partial y_i^n}{\partial z_{i,(j,k)}^l}), 0\}, \quad (3)$$

where $i$ is the index of the $i$-th sample; $j \in [1, 2, ..., h = \frac{H}{u}]$, $h$ is the number obtained through dividing the image height $H$ by the local block height $u$; $k \in [1, 2, ..., w = \frac{W}{u}]$, $w$ is the number obtained through dividing the image width $W$ by the local block width $u$; the patch size $u$ of square block is set to $\max (\min (H, W) / 32, 8)$; $l$ is the index of the feature space dimension $d$, $z^l$ is the $l$-th dimensional feature of the transitional output $z_i$; $n$ is the class index and $y^n$ is the classification prediction for the $n$-th class. $\frac{\partial y_i^n}{\partial z_{i,(j,k)}^l}$ is the gradient information obtained by backpropagation of $n$-th class on the $l$-th dimensional feature. The operation of $max\{Tanh(\cdot)\}$ is employed to suppress the negative pixels belonging to other categories that the model does not focus on. In the absence of $max\{Tanh(\cdot)\}$, the weight values sometimes do not effectively highlight the target class alone, leading to worse performance in feature localization. Next, FSG retains the areas with model-interested features and replaces the areas that the model is not interested in with images generated by the diffusion model. By image fusion, target data with feature differences can be expressed as:

$$\widetilde{x}_i = x_i[weight_{i,j,k} > r_i] \odot x_{i,(g)}[weight_{i,j,k} < r_i], \tag{4}$$

where $r_i$ is controlled by hyperparameter $r$ is to determine the ratio between areas of model interest and non-interest, $r_i = \frac{r}{h \times w} \sum_{j=1}^{h} \sum_{k=1}^{w} weight_{i,j,k}$; $\odot$ is the element-wise multiplication symbol. $x_{i,(g)}$ represents the $i$-th generated data output by the diffusion model; the generated fusion domain is defined as $\widetilde{D}_t = \{(\widetilde{x}_i)\}_{i=1}^{N_t}$.

**Fine-tuning Multi-modal Model.** To better align the diffusion model with the target domain style, RrED fine-tunes the text encoder from the diffusion model to rectify reasoning errors that arise during the inference process. Before fine-tuning, we introduce the diffusion-based predictor to leverage reliable semantic information from the diffusion model. Meanwhile, we introduce a matching image encoder from [20] to fine-tune the text encoder. In fine-tuning, we introduce the prompt learning to adapt BUDA by inserting learnable continuous vectors into the original text input, as follows:

$$Prompt\ Text = \{[v_1], ..., [v_{\frac{m}{2}}], [CLS], [v_{\frac{m}{2}+1}], ..., [v_m]\}, \tag{5}$$

where $[v_1], ..., [v_m]$ denote prompt word embeddings of the same dimensionality, $m$ is the number of context tokens, and $[CLS]$ is the class name. Then, we propose a task-specific prompt learning loss to accomplish the fine-tuning of multi-modal model, which can be expressed as:

$$\mathcal{L}_{\mathcal{V}_\theta} = -\min_{\mathcal{V}_\theta} \mathbb{E}_{x_i, \widetilde{x}_i \in D_t, \widetilde{D}_t} (S(x_i)\ or\ p_\theta(x_i))^T \log \mathcal{V}_\theta(\widetilde{x}_i), \tag{6}$$

where $\mathcal{V}_\theta$ is defined as the multi-modal model; during the training period, only prompt word embeddings $[v_1], ..., [v_m]$ are unlocked, while all other parameters are fixed; $p_\theta(x_i)$ is diffusion model prediction from the diffusion-based predictor; $S(x_i)$ is the ALS prediction for the $i$-th sample used as input from the smooth label repository and used for the initialization of the multi-modal model. *As shown in Appendix G*, when only source domain knowledge is available, domain discrepancy causes the model to fail in adapting well to the target domain. The diffusion model has more reliable semantic knowledge than the black-box predictor. Therefore, after initialization, $S(x_i)$ in the fine-tuning process is replaced by $p_\theta(x_i)$.

**Adaptation Loss in DTR.** During the early and middle stages of training, domain discrepancies often result in noisy and unreliable pseudo-labels for the target domain. Such discrepancies amplify erroneous gradients throughout the training process, thereby heightening the probability of compromised feature learning and detrimental knowledge transfer [19]. Therefore, we propose a guidance correction to use the diffusion model rich in semantic knowledge to guide the learning of the target model, which can be expressed as:

$$\mathcal{L}_{GC} = -\min_{\mathcal{M}_\theta} \mathbb{E}_{x_i, \widetilde{x}_i \in D_t, \widetilde{D}_t} p_\theta(x_i)^T \log \mathcal{M}_\theta(\widetilde{x}_i), \tag{7}$$

where $\mathcal{L}_{GC}$ is a standard cross-entropy function. The entropy minimization process defined in $\mathcal{L}_{GC}$ inherently biases predictions toward sample-dense areas in the feature distribution, consequently diminishing model generalization ability [46]. To mitigate this effect, we introduce a conditional constraint loss to limit this impact while maintaining robust feature space consolidation:

$$\mathcal{L}_{CC} = -\min_{\mathcal{M}_\theta} \sum_{i=1}^{N_t} \sum_{j=1}^{N_t} (\mathbf{I} - \frac{tr\{\mathcal{M}_\theta(\widetilde{x}_i)^T \mathcal{M}_\theta(\widetilde{x}_j)\}}{\|\mathcal{M}_\theta(\widetilde{x}_i)\| \|\mathcal{M}_\theta(\widetilde{x}_j)\|}) \cdot \mathcal{M}_\theta(\widetilde{x}_i)^T \mathcal{M}_\theta(\widetilde{x}_j), \tag{8}$$

Table 2: Accuracies (%) on the *Office-Home* using ResNet-50 and the *VisDA-17* using ResNet-101. The setting of **U**, **SF**, and **BP** corresponds to UDA, SFDA, and BUDA, respectively. **P** and **D** indicate whether external prompts and diffusion model are utilized (✓) or not (×). "Source-only" refers to using the black-box predictor to evaluate the predicted target samples. The top-performing BUDA methods are highlighted in bold. *The complete results on VisDA-17 are in Appendix B.*

| Method | Setting | P | D | A→C | A→P | A→R | C→A | C→P | C→R | P→A | P→C | P→R | R→A | R→C | R→P | Mean | VisDA Mean |
|---|---|---|---|---|---|---|---|---|---|---|---|---|---|---|---|---|---|
| Source-only | – | × | × | 44.1 | 66.9 | 74.2 | 54.5 | 63.3 | 66.1 | 52.8 | 41.2 | 73.2 | 66.1 | 46.7 | 77.5 | 60.6 | 48.9 |
| HMA [47] | U | × | × | 60.6 | 79.1 | 82.9 | 68.9 | 77.5 | 79.3 | 69.1 | 55.9 | 83.5 | 74.6 | 62.3 | 84.4 | 73.2 | 88.1 |
| DAPL [48] | U | ✓ | × | 54.1 | 84.3 | 84.4 | 74.4 | 83.7 | 85.0 | 74.5 | 54.6 | 84.8 | 75.2 | 54.7 | 83.8 | 74.5 | 86.9 |
| PDA [49] | U | ✓ | × | 55.4 | 85.1 | 85.8 | 75.2 | 85.2 | 85.2 | 74.2 | 55.2 | 85.8 | 74.7 | 55.8 | 86.3 | 75.3 | 89.7 |
| DATUM [50] | U | ✓ | ✓ | 49.3 | 68.4 | 72.8 | 70.6 | 69.3 | 72.1 | 69.9 | 50.2 | 73.9 | 77.1 | 51.5 | 75.8 | 66.7 | 75.3 |
| S-Fusion [26] | U | ✓ | ✓ | 57.4 | 76.0 | 80.2 | 67.7 | 76.5 | 77.6 | 67.9 | 56.4 | 81.2 | 75.6 | 62.1 | 86.4 | 72.1 | 86.1 |
| DACDM [24] | U | ✓ | ✓ | 60.4 | 78.8 | 82.7 | 69.6 | 80.5 | 79.6 | 65.2 | 58.3 | 83.1 | 75.8 | 64.2 | 85.6 | 73.6 | 86.8 |
| DAD [25] | U | ✓ | ✓ | 62.5 | 78.6 | 83.0 | 70.4 | 79.2 | 79.8 | 70.2 | 58.3 | 83.1 | 76.3 | 63.5 | 88.2 | 74.4 | 90.0 |
| PLUE [51] | SF | × | × | 49.1 | 73.5 | 78.2 | 62.9 | 73.5 | 74.5 | 62.2 | 48.3 | 78.6 | 68.6 | 51.8 | 81.5 | 66.9 | 90.0 |
| C-SFDA [10] | SF | × | × | 58.6 | 80.2 | 82.9 | 69.8 | 78.6 | 79.0 | 67.8 | 55.7 | 82.3 | 73.6 | 60.1 | 84.9 | 72.8 | 87.8 |
| DIFO [52] | SF | ✓ | × | 70.6 | 90.6 | 88.8 | 82.5 | 90.6 | 88.8 | 80.9 | 70.1 | 88.9 | 83.4 | 70.5 | 91.2 | 83.1 | 90.3 |
| DINE [15] | BP | × | × | 52.2 | 78.4 | 81.3 | 65.3 | 76.6 | 78.7 | 62.7 | 49.6 | 82.2 | 69.8 | 55.8 | 84.2 | 69.7 | 75.6 |
| BiMem [40] | BP | × | × | 54.5 | 78.8 | 81.4 | 66.7 | 78.7 | 79.6 | 65.9 | 53.6 | 82.3 | 73.6 | 57.8 | 84.9 | 71.5 | 83.6 |
| BETA [16] | BP | × | × | 57.2 | 78.5 | 82.1 | 68.0 | 78.6 | 79.7 | 67.5 | 56.0 | 83.0 | 71.9 | 58.9 | 84.2 | 72.1 | 85.1 |
| RFC [18] | BP | × | × | 57.4 | 80.0 | 82.8 | 67.0 | 80.6 | 80.2 | 68.3 | 57.8 | 82.8 | 72.8 | 59.3 | 85.9 | 72.9 | 85.2 |
| SEAL [17] | BP | × | × | 58.5 | 81.4 | 84.7 | 71.7 | 80.4 | 82.1 | 72.2 | 54.3 | 86.0 | 76.2 | 60.6 | 86.3 | 74.5 | 89.2 |
| AEM [19] | BP | ✓ | × | 65.4 | 88.3 | 89.5 | 80.1 | 90.7 | 89.7 | 78.9 | 61.4 | 89.9 | 79.2 | 63.6 | 90.8 | 80.6 | 89.3 |
| RrED | BP | ✓ | ✓ | **82.3** | **93.9** | **90.0** | **82.0** | **93.7** | **90.1** | **82.6** | **83.0** | **90.4** | **84.7** | **83.3** | **94.1** | **87.5** | **91.2** |

where $tr\{\cdot\}$ is the trace of a matrix. The prediction discriminability within a mini-batch reaches its peak when the lower bound of $\mathcal{L}_{CC}$ and the minimum of $\mathcal{L}_{GC}$ are attained simultaneously, yielding fully determined prediction matrices. In DTR, the objective loss for the target model can be expressed as:

$$\mathcal{L}_{\mathcal{M}_\theta(DTR)} = \mathcal{L}_{task} + \mathcal{L}_{GC} + \gamma\mathcal{L}_{CC}, \tag{9}$$

where $\gamma$ is a hyperparameter to control the role of the conditional constraint loss $\mathcal{L}_{CC}$.

### 3.3 Self-Rectifying Reasoning Model

**Sample-Version Interaction (SVI).** Before the second phase begins, we substitute the original text encoder in the diffusion model for the fine-tuned text encoder with prompt word embeddings. In this regard, diffusion can generate images with more stable target domain styles. Then, we generate differentiated synthetic images according to the process of FSG, and perform interactive learning between the synthetic images and the target images in SVI. For this, our proposed interactive learning can be divided into two parts: (1) the former term corrects model reasoning errors by measuring similarity between different versions of predictions, enforcing scattered data distribution boundaries to stabilize around the nearest feature cluster centers; (2) the latter term strengthens feature discrepancies between different versions of predictions to enhance model generalization and prevent overfitting. The interactive optimization can be expressed as:

$$\mathcal{L}_{SVI} = -\min_{\mathcal{M}_\theta}\mathbb{E}_{x_i,\widetilde{x}_i \in D_t, \widetilde{D}_t} \underbrace{w\log\{sim\left(\mathcal{M}_\theta(x_i), \mathcal{M}_\theta(\widetilde{x}_i)\right)\}}_{\text{Rectify reasoning errors}} - \underbrace{\log\{1 - sim\left(\mathcal{M}_\theta(x_i), \mathcal{M}_\theta(\widetilde{x}_i)\right)\}}_{\text{Strengthen feature discrepancies}},$$
$$\tag{10}$$

where $sim(\cdot)$ denotes the operation of calculating cosine similarity; $w$ is employed to assign different weights according to the similarities between the features in the target data predictions $\mathcal{M}_\theta(x_i)$ and the synthesized data predictions $\mathcal{M}_\theta(\widetilde{x}_i)$. The similarity weight $w$ can be formulated as:

$$w = exp(-sort(-\log\{sim\left(\mathcal{M}_\theta(x_i), \mathcal{M}_\theta(\widetilde{x}_i)\right)\})), \tag{11}$$

where $sort(\cdot)$ denotes sorting in descending order and returning the corresponding indices of the samples.

**Adaptation Loss in SRM.** In the SRM stage, our goal is to enable self-rectification of the target model by contrasting the reasoning discrepancies among different versions of the same sample, thereby enhancing its reasoning ability for better adaptation to the target domain. In SRM, the objective loss for the target model can be expressed as:

$$\mathcal{L}_{\mathcal{M}_\theta(SRM)} = \mathcal{L}_{task} + \mathcal{L}_{SVI}. \tag{12}$$

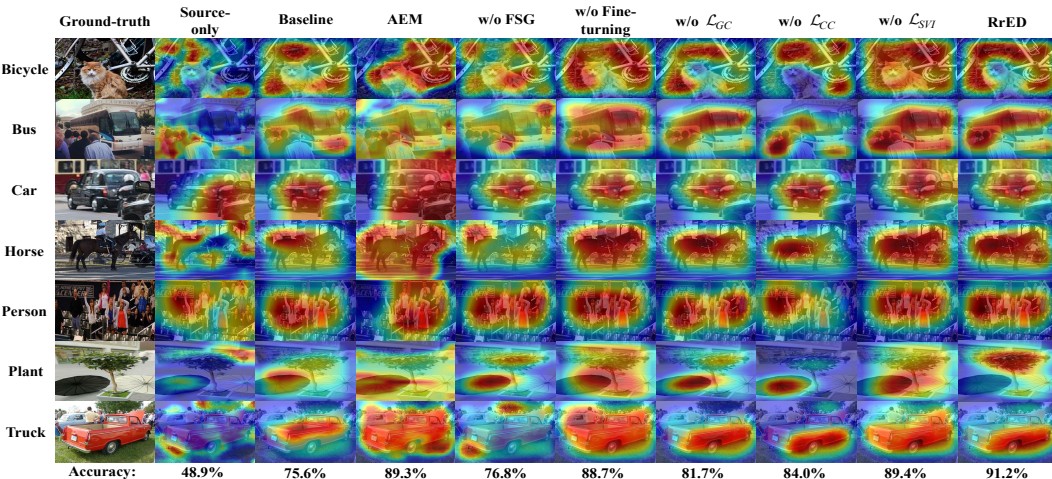

| | Ground-truth | Source-only | Baseline | AEM | w/o FSG | w/o Fine-turning | w/o $\mathcal{L}_{GC}$ | w/o $\mathcal{L}_{CC}$ | w/o $\mathcal{L}_{SVI}$ | RrED |
|---|---|---|---|---|---|---|---|---|---|---|
| Accuracy: | | 48.9% | 75.6% | 89.3% | 76.8% | 88.7% | 81.7% | 84.0% | 89.4% | 91.2% |

Figure 3: Qualitative and quantitative ablation studies on *VisDA-17* using Grad-CAM [45]. Each result is reported when the best accuracy is achieved. Zooming for a clearer view. *The complete quantitative results of ablation are reported in Appendix E.*

To explain why our algorithm RrED works effectively and why it contributes to BUDA, *we derive an error bound through theoretical analysis in Appendix C. Moreover, the whole training process of RrED is shown in Appendix D.*

## 4 Experiments

**Datasets.** RrED is evaluated on four widely-used domain adaptation benchmarks. *Office-31* [53] is a small-scale dataset with 4,110 images in 31 categories from three domains: Amazon (A), Dslr (D), and Webcam (W). *Office-Home* [54] is a medium-scale dataset, containing 15.5K images across 65 categories from four domains: Real World (R), Clipart (C), Art (A), and Product (P). *VisDA-17* [55] is a large-scale benchmark, including 152K synthetic images (source) and 55K real-world images (target) across 12 categories, emphasizing the synthetic-to-real domain gap. *DomainNet* [56] is the most extensive benchmark, with about 600K images. Following previous methods [17, 52], the evaluation setup for adaptation scenarios involves merely 4 domains with 126 categories, including

Table 3: Accuracies (%) on the *Office-31* using ResNet-50 backbone.

| Method | Setting | P | D | A→D | A→W | D→A | D→W | W→A | W→D | Mean |
|---|---|---|---|---|---|---|---|---|---|---|
| Source-only | – | × | × | 79.9 | 76.6 | 56.4 | 92.8 | 60.9 | 98.5 | 77.5 |
| HMA [47] | U | × | × | 95.8 | 95.1 | 79.3 | 99.3 | 77.6 | 100 | 91.2 |
| DAPL [48] | U | ✓ | × | 81.7 | 80.3 | 81.2 | 81.8 | 81.0 | 81.3 | 81.2 |
| PDA [49] | U | ✓ | × | 91.2 | 92.1 | 83.5 | 98.1 | 82.5 | 99.8 | 91.2 |
| DATUM [50] | U | ✓ | ✓ | 89.3 | 83.7 | 80.5 | 88.4 | 81.7 | 97.3 | 86.8 |
| S-Fusion [26] | U | ✓ | ✓ | 94.8 | 95.3 | 78.3 | 99.1 | 78.6 | 100 | 91.0 |
| DACDM [24] | U | ✓ | ✓ | 97.5 | 96.9 | 79.8 | 98.9 | 77.7 | 97.5 | 91.8 |
| DAD [25] | U | ✓ | ✓ | 95.6 | 98.5 | 81.4 | 99.5 | 82.2 | 100 | 92.8 |
| PLUE [51] | SF | × | × | 89.2 | 88.4 | 72.8 | 97.1 | 69.6 | 97.9 | 85.8 |
| C-SFDA [10] | SF | × | × | 96.2 | 93.9 | 77.3 | 98.8 | 77.9 | 99.7 | 90.5 |
| SF(DA)² [9] | SF | × | × | 95.8 | 92.1 | 75.7 | 99.0 | 76.8 | 99.8 | 89.9 |
| DIFO [52] | SF | ✓ | × | 97.2 | 95.5 | 83.0 | 97.2 | 83.2 | 98.8 | 92.5 |
| DINE [15] | BP | × | × | 91.6 | 86.8 | 72.2 | 96.2 | 73.3 | 98.6 | 86.4 |
| BiMem [40] | BP | × | × | 92.8 | 88.2 | 73.9 | 96.8 | 75.3 | 99.4 | 87.7 |
| BETA [16] | BP | × | × | 93.6 | 88.3 | 76.1 | 95.5 | 76.5 | 99.0 | 88.2 |
| RFC [18] | BP | × | × | 94.4 | 93.0 | 76.7 | 95.6 | 77.5 | 98.1 | 89.2 |
| SEAL [17] | BP | × | × | 95.1 | 88.3 | 77.6 | 96.0 | 76.7 | 99.3 | 88.8 |
| AEM [19] | BP | ✓ | × | 95.1 | 94.0 | 81.8 | 98.2 | 82.6 | 99.4 | 91.9 |
| RrED | BP | ✓ | ✓ | **97.8** | **95.9** | **83.7** | **99.1** | **84.5** | **99.8** | **93.5** |

Real (R), Clipart (C), Painting (P), and Sketch (S). There is a need to overcome the domain gaps among 12 subtasks with different adaptation scenarios.

**Comparison Methods.** We evaluate the performance of RrED by comparing it with several related methods across UDA, SFDA, and BUDA settings. For UDA, we conduct comparisons with HMA [47], DAPL [48], DATUM [50], S-Fusion [26], DACDM [24], DAD [25], PDA [49], and AD-CLIP [57]. For SFDA, we compare with PLUE [51], C-SFDA [10], SF(DA)² [9], TPDS [58], and DIFO [52]. For BUDA, we compare with previous SOTA methods, including DINE [15], BiMem [40], BETA [16], RFC [18], SEAL [17], and AEM [19]. Among them, the previous methods include DATUM, S-Fusion, DACDM, and DAD use the diffusion model as an external prompt; DAPL, PDA,

Table 4: Accuracies (%) on the *DomainNet* using ResNet-50 backbone.

| Method | Setting | P | D | C→P | C→R | C→S | P→C | P→R | P→S | R→C | R→P | R→S | S→C | S→P | S→R | Mean |
|---|---|---|---|---|---|---|---|---|---|---|---|---|---|---|---|---|
| Source-only | − | × | × | 36.1 | 52.1 | 41.3 | 40.7 | 56.5 | 34.6 | 48.3 | 46.8 | 35.2 | 50.5 | 35.9 | 46.1 | 43.7 |
| DAPL [48] | U | ✓ | × | 72.4 | 87.6 | 65.9 | 72.7 | 87.6 | 65.6 | 73.2 | 72.4 | 66.2 | 73.8 | 72.9 | 87.8 | 74.8 |
| AD-CLIP [57] | U | ✓ | × | 71.7 | 88.1 | 66.0 | 73.2 | 86.9 | 65.2 | 73.6 | 73.0 | 68.4 | 72.3 | 74.2 | 89.3 | 75.2 |
| PLUE [51] | SF | × | × | 59.8 | 74.0 | 56.0 | 61.6 | 78.5 | 57.9 | 61.6 | 65.9 | 53.8 | 67.5 | 64.3 | 76.0 | 64.7 |
| TPDS [58] | SF | × | × | 62.9 | 77.1 | 59.8 | 65.6 | 79.0 | 61.5 | 66.4 | 67.0 | 58.2 | 68.6 | 64.3 | 75.3 | 67.1 |
| DIFO [52] | SF | ✓ | × | 76.6 | 87.2 | 74.9 | 80.0 | 87.4 | 75.6 | 80.8 | 77.3 | 75.5 | 80.5 | 76.7 | 87.3 | 80.0 |
| DINE [15] | BP | × | × | 43.7 | 61.5 | 44.0 | 44.0 | 62.9 | 38.7 | 54.3 | 53.1 | 41.7 | 54.0 | 44.5 | 59.3 | 50.1 |
| BETA [16] | BP | × | × | 48.3 | 64.7 | 49.2 | 49.6 | 66.3 | 43.4 | 58.1 | 57.7 | 45.7 | 58.7 | 49.9 | 63.1 | 54.5 |
| SEAL [17] | BP | × | × | 49.5 | 67.9 | 48.7 | 49.9 | 68.5 | 44.0 | 60.6 | 57.4 | 46.7 | 59.2 | 50.4 | 67.1 | 55.8 |
| AEM [19] | BP | ✓ | × | 66.4 | 77.8 | **72.1** | 80.0 | 86.7 | 69.1 | 79.5 | 76.6 | 67.8 | 78.1 | 72.6 | 77.6 | 75.4 |
| RrED | BP | ✓ | ✓ | **76.8** | **87.9** | 71.9 | **81.5** | **88.7** | 74.7 | **83.5** | **80.1** | 73.0 | **81.7** | **78.3** | **88.3** | **80.5** |

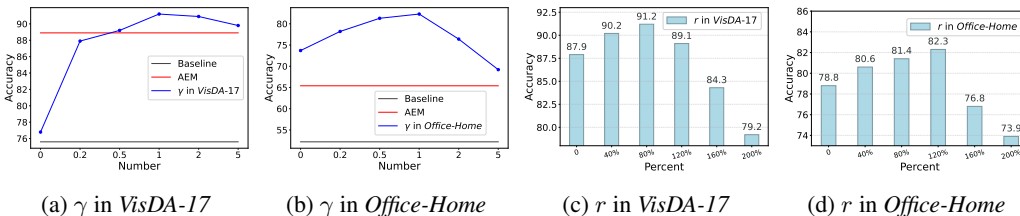

(a) $\gamma$ in *VisDA-17*    (b) $\gamma$ in *Office-Home*    (c) $r$ in *VisDA-17*    (d) $r$ in *Office-Home*

Figure 4: The accuracy trends of predictions on the *VisDA-17* and *Office-Home* (A→C). $\gamma$ controls the effect of $\mathcal{L}_{CC}$, as shown in (a) and (b). $r$ determines the ratio between regions of interest and non-interest, as shown in (c) and (d).

AD-CLIP, DIFO, and AEM use the multi-modal model CLIP [20] as an external prompt. *Specific implementation details are shown in Appendix F.*

**Results.** As reported in Tables 2, 3, and 4, RrED achieves consistently superior performance over previous SOTA methods across all four benchmarks. We choose DINE [15] as the baseline. In terms of average accuracy, RrED surpasses the prior BUDA approach AEM [19] by 6.9%, 1.9%, 1.6%, and 5.1% on the *Office-Home*, *VisDA-17*, *Office-31*, and *DomainNet*, respectively. Furthermore, compared to the diffusion-based UDA method DAD [25], RrED achieves a maximum improvement of 13.1% on the *Office-Home*. These results demonstrate, by introducing the diffusion model into black-box learning through a two-stage strategy and fine-tuning the diffusion model's text encoder, RrED more effectively utilizes the diffusion model to enhance target model discriminability compared to previous methods that either generate semantically-rich additional samples or directly perform prediction using diffusion model. Furthermore, RrED exhibits significantly superior performance in scenarios with more stringent security protection constraints compared to previous diffusion-based DA methods.

**Ablation Study.** Figure 3 presents our ablation studies on the *VisDA-17* using Grad-CAM [45] visualizations. RrED is inspired by human decision-making systems and aims to refine model reasoning through a two-stage correction process. In computer vision, the model's reasoning process mirrors human decision-making by utilizing existing knowledge and feature similarity computations to determine whether targets in complex scenes meet the task requirements. As indicated in Figure 2, RrED's two-stage correction focuses on improving the diffusion model's reasoning ability, thereby guiding and enhancing the reasoning of the target model. Therefore, verifying whether the model's reasoning ability improves during optimization is a central focus of our experiments. To verify the model's reasoning ability, we introduce Grad-CAM, which is well-known for validating the reasoning ability of models. For the functional, we employ Grad-CAM in Figure 3 to highlight the vital and irreplaceable roles play in the overall performance. As shown in Figure 3, Grad-CAM clearly demonstrates that our model exhibits stronger reasoning capabilities, better object recognition, and more precise capture of fine-grained features in target samples compared with existing SOTA methods. In DTR period, FSG is designed to prevent the diffusion-generated images from causing irreversible negative effects. When FSG is removed, the uncontrolled images solely generated by the diffusion model mislead the target model, resulting in a significant performance drop. $\mathcal{L}_{GC}$ and $\mathcal{L}_{CC}$ act in a complementary manner: $\mathcal{L}_{GC}$ enhances the model's discriminative capability via cross-entropy learning, while $\mathcal{L}_{CC}$ mitigates the sample enrichment effect introduced by $\mathcal{L}_{GC}$ to improve model's generalization. When $\mathcal{L}_{CC}$ is removed, the target model exhibits the overfitting phenomenon prematurely. Only when both components work jointly can the full effectiveness be

realized. In SRM period, $\mathcal{L}_{SVI}$ performs self-rectifying inference learning by integrating interactive learning with the samples generated by the diffusion model whose text encoder has been fine-tuned. The combination of $\mathcal{L}_{SVI}$ with the fine-tuned diffusion model allows the overall model to capture key features more accurately. From the perspective of model reasoning, we observe from Figure 3 that (1) FSG enhances generalization and helps the model attend to the correct class-discriminative regions; (2) $\mathcal{L}_{GC}$ and $\mathcal{L}_{CC}$ jointly determine the approximate region of feature extraction from the target model; (3) the fine-tuned text encoder and $\mathcal{L}_{SVI}$ jointly optimize features for the region of interest of the model. *More ablation studies are shown in Appendix E.*

**Parameter Analysis and Comparison.** As shown in Figure 4, the effects under different values of $\gamma$ and $r$ are presented. $\gamma$ controls $\mathcal{L}_{CC}$ to modulate the distributional density of samples within the feature space. When $\lambda$ is equal to 0, $\mathcal{L}_{CC}$ is not effective; when $\lambda$ is equal to 5, excessive amplification of feature discrepancies severely impairs the model's ability to distinguish between samples. For the large-scale dataset *VisDA-17*, appropriate $\lambda$ leads to notable improvements. For the medium-scale *Office-Home*, the differences between samples more significantly affect the model's discriminative ability. $r$ determines the ratio between regions of interest and non-interest. When $r$ is 0, FSG outputs the original target domain samples. When $r$ is 200%, FSG fails, and the outputs are entirely generated by the diffusion model. These results indicate that selecting an appropriate $r$ can effectively enhance the generalization ability of the target model, while also demonstrating that images generated solely by the diffusion model are unreliable. *More visual comparisons and further analysis are provided in Appendix G. The computational consumption is presented in Appendix H.*

## 5  Conclusion

In this paper, we observe that existing methods have certain weaknesses. To tackle them, we propose a diffusion-based algorithm, RrED, the first to introduce the diffusion model into the BUDA task and perform task-specific fine-tuning on its text encoder. Inspired by the human decision-making process, RrED is composed of DTR and SRM stages: DTR facilitates the target model's training by correcting reasoning errors from the diffusion model while harnessing its implicit knowledge; SRM refines the learning process by utilizing the target model's differential reasoning in combination with samples produced by the fine-tuned diffusion model. The experimental results show that RrED enhances class discrimination ability and model reasoning ability, ultimately achieving performance improvements far exceeding previous SOTA methods on all evaluated datasets.

## Acknowledgment

This work was supported in part by the National Natural Science Foundation of China under Grant 62176162 and in part by Guangdong Basic and Applied Basic Research Foundation under Grant 2023A1515012875 and Grant 2022A1515140099.

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

# Appendix

## A. Diffusion Process for BUDA

A standard diffusion model (e.g., DDPM [21]) consists of two core components: a forward diffusion operator $q$ and a reverse denoising operator $p$. In the forward process, DDPM diffuses the target data distribution by gradually injecting Gaussian noise into the data point $x_i$ over the total $ST$ steps using a fixed Markov chain. The forward diffusion operator $q$ can be expressed as:

$$q(x_i^j | x_i^{j-1}) = \mathcal{N}(x_i^j; \sqrt{1 - \beta_j} x_i^{j-1}, \beta_j \mathbf{I}), \tag{13}$$

where $j$ is defined as the diffusion timestep, $j \in [0, 1, ..., ST]$; $\beta_j$ is the fixed variance scheduler that controls the scale of the Gaussian noise, $\beta_j \in [0, 1]$; $\mathcal{N}$ represents the probability density function of the Gaussian distribution; $\mathbf{I}$ is a unit vector; and $\beta_j \mathbf{I}$ is the covariance matrix. In the reverse denoising operator, DDPM concentrates the target data distribution by gradually generating a sequence of denoised images $x_g$ over the same $T$ steps. The reverse denoising operator $p$ can be expressed as:

$$p(x_g^{k-1} | x_g^k) = \mathcal{N}(x_g^{k-1}; \frac{1}{\sqrt{\alpha_k}}(x_g^k - \frac{1 - \alpha_k}{\sqrt{1 - \overline{\alpha}_k}} \sigma_\theta(x_g^k, k)), \beta_k \mathbf{I}), \tag{14}$$

where $k$ is defined as the denoising timestep, $k \in [ST, ST - 1, ..., 0]$; $\alpha_k = 1 - \beta_k$; $\overline{\alpha}_i = \prod_{i=1}^k \alpha_i$; and $\sigma_\theta(x_g^k, k)$ predicts the noise at the current timestep $k$ and denoises the corresponding input data $x_g^k$, $\sigma_\theta(x_g^k, k) \in [0, 1]$. To introduce the diffusion model into BUDA, we follow [42] to add a predictor $p_\theta$ with fixed weights to judge the category of the target data $x_i$. The judgment process of the predictor is as follows:

$$p_\theta(L_j | x_i) = \frac{\exp(-\mathbb{E}_{k \in T} \| \sigma - \sigma_\theta(x_i^k, L_j) \|^2)}{\sum_{l=1}^b \exp(-\mathbb{E}_{k \in T} \| \sigma - \sigma_\theta(x_i^k, L_l) \|^2)}, \tag{15}$$

where $L_j$ is a low-dimensional text embedding corresponding to the $j$-th class of the $i$-th sample $x_i$; $b$ is the number of classes; $\sigma$ follows the standard Gaussian distribution $\mathcal{N}(0, 1)$. In this work, we introduce an image encoder with fixed weights from CLIP [20] and leverage the predictor along with the image encoder during the diffusion-target model rectification to fine-tune the text encoder. Moreover, before the SRM period, the fine-tuned text encoder is fed back to the diffusion model to enable more controllable image generation and adapt the target model to the BUDA task.

## B. Supplement of Complete Experimental Results

As shown in Table 5, the comparison results demonstrate that our RrED effectively employs a two-stage strategy guided by diffusion model for target model optimization, achieving significantly

Table 5: The complete accuracies (%) on the *VisDA-17* using ResNet-101 backbone.

| Method | Setting | P | D | plane | bike | bus | car | horse | knife | mcycle | person | plant | sktbrd | train | truck | Mean |
|---|---|---|---|---|---|---|---|---|---|---|---|---|---|---|---|---|
| Source-only | − | × | × | 64.3 | 24.6 | 47.9 | 75.3 | 69.6 | 8.5 | 79.0 | 31.6 | 64.4 | 31.0 | 81.4 | 9.2 | 48.9 |
| HMA [47] | U | × | × | 97.6 | 88.4 | 84.3 | 76.0 | 98.4 | 97.1 | 91.3 | 81.4 | 97.0 | 96.7 | 88.8 | 60.7 | 88.1 |
| DAPL [48] | U | ✓ | × | 97.8 | 83.1 | 88.8 | 77.9 | 97.4 | 91.5 | 94.2 | 79.7 | 88.6 | 89.3 | 92.5 | 62.0 | 86.9 |
| PDA [49] | U | ✓ | × | 99.2 | 91.1 | 91.9 | 77.1 | 98.4 | 93.6 | 95.1 | 84.9 | 87.2 | 97.3 | 95.3 | 65.3 | 89.7 |
| DATUM [50] | U | ✓ | ✓ | 85.7 | 76.4 | 79.7 | 75.4 | 84.1 | 82.3 | 80.4 | 76.7 | 81.9 | 82.6 | 78.4 | 20.2 | 75.3 |
| S-Fusion [26] | U | ✓ | ✓ | 92.9 | 83.7 | 89.3 | 87.0 | 95.3 | 92.7 | 90.1 | 86.8 | 92.2 | 93.2 | 88.3 | 42.0 | 86.1 |
| DACDM [24] | U | ✓ | ✓ | 96.2 | 84.8 | 83.2 | 73.3 | 94.8 | 96.6 | 91.0 | 88.2 | 93.0 | 93.4 | 87.5 | 59.7 | 86.8 |
| DAD [25] | U | ✓ | ✓ | 97.4 | 89.6 | 92.2 | 91.6 | 97.3 | 97.0 | 95.1 | 89.8 | 97.2 | 96.9 | 93.7 | 42.5 | 90.0 |
| PLUE [51] | SF | × | × | 97.3 | 96.2 | 90.5 | 91.8 | 90.0 | 94.2 | 87.4 | 87.7 | 97.0 | 84.3 | 93.0 | 81.0 | 90.0 |
| C-SFDA [10] | SF | × | × | 97.6 | 88.8 | 86.1 | 72.2 | 97.2 | 94.4 | 92.1 | 84.7 | 93.0 | 90.7 | 93.1 | 63.5 | 87.8 |
| SF(DA)$^2$ [9] | SF | × | × | 96.8 | 89.3 | 82.9 | 81.4 | 96.8 | 95.7 | 90.4 | 81.3 | 95.5 | 93.7 | 88.5 | 64.7 | 88.1 |
| DIFO [52] | SF | ✓ | × | 97.7 | 87.6 | 90.5 | 83.6 | 96.7 | 95.8 | 94.8 | 74.1 | 92.4 | 93.8 | 92.9 | 65.5 | 88.8 |
| DINE [15] | BP | × | × | 81.4 | 86.7 | 77.9 | 55.1 | 92.2 | 34.6 | 80.8 | 79.9 | 87.3 | 87.9 | 84.3 | 58.7 | 75.6 |
| BETA [16] | BP | × | × | 94.9 | 90.2 | 85.4 | 61.1 | 95.5 | 93.1 | 85.0 | 83.8 | 92.9 | 91.9 | 91.1 | 55.0 | 85.1 |
| RFC [18] | BP | × | × | 95.6 | 89.7 | 87.8 | 75.8 | 96.5 | 96.5 | 90.4 | 82.8 | 96.0 | 70.0 | 85.7 | 55.1 | 85.2 |
| SEAL [17] | BP | × | × | 97.9 | **92.2** | 88.0 | 73.5 | 97.1 | 96.1 | 92.4 | 85.7 | 93.9 | 95.6 | 91.2 | **66.4** | 89.2 |
| AEM [19] | BP | ✓ | × | **98.6** | 88.1 | **89.7** | 74.8 | 98.0 | 93.9 | 93.0 | **89.3** | 90.1 | **97.2** | **95.2** | 63.5 | 89.3 |
| RrED | BP | ✓ | ✓ | 97.5 | 91.9 | 88.1 | **88.0** | **98.1** | **96.9** | **94.3** | 88.8 | **96.6** | 96.6 | 94.1 | 63.8 | **91.2** |

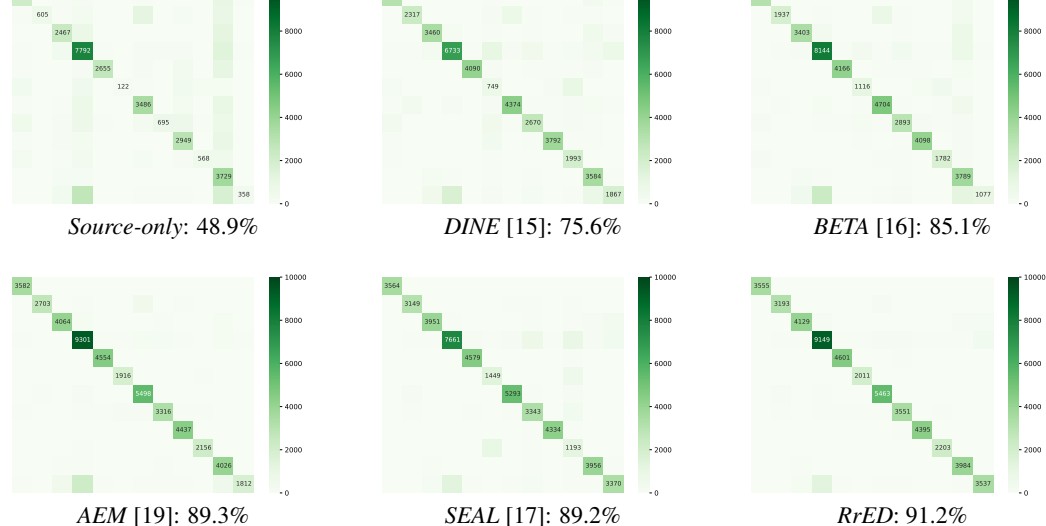

Figure 5: Classification results on *VisDA-17* are visualized with a confusion matrix. Note that all these results are obtained through the evaluation which is conducted in the same experimental environment. (Zooming in for a clear view)

greater improvements on the large-scale benchmark *VisDA-17* [55]. Furthermore, we observe that RrED does not outperform some comparison methods [17, 19] on certain classes. We attribute this to the fact that our target model is trained under the guidance of a diffusion model, which tends to focus on broad class distinctions to enhance overall discriminative ability. In contrast, the distillation-based method SEAL [17] exhibits slight overfitting to a few specific classes (e.g., the "bike" and "truck" class), resulting in higher recognition accuracy on those classes but reduced performance on others. The CLIP-based method AEM [19] demonstrates notable discriminative power on specific classes. Based on our analysis of the CLIP model [20], we find that these classes are often overrepresented during CLIP pretraining. For example, there is a strong similarity between the "person" class in the target domain and pretraining classes such as "baseball player", "bridegroom", and "scuba diver" in CLIP model. In contrast, classes that are absent (e.g., the "knife" class) or rarely seen (e.g., the "plant" class) during pretraining tend to have much lower recognition accuracy. In addition, we supplement the classification visualization of the *VisDA-17* in Figure 5. For a fair comparison, all the classification visualizations are obtained in the same experimental environment. These results highlight that our RrED method substantially surpasses other BUDA approaches in improving class discrimination ability.

## C. Theoretical Justifications

We provide theoretical justifications grounded in the generalization bound of reasoning to clarify the working mechanism of our algorithm.

First, we adopt PAC-Bayes theory [59] for the classification task to optimize the target model with the uncertainty estimation of the black-box predictor.

**Theorem 1** [60]. Given a target data distribution $D_t$, a hypothesis $H$, and a prior distribution $\pi$ over the hypothesis space $\Theta$. For any $\tau \in (0, 1]$ and $\lambda > 0$, with a probability at least $1 - \tau$ over the target samples $x_t \sim D_t$, for all posteriors $\rho$, we have:

$$\mathbb{E}_{\rho(H)}\left[\mathcal{L}(H)\right] \leq \mathbb{E}_{\rho(H)}\left[\tilde{\mathcal{L}}_{x_t}(H)\right] + \frac{1}{\lambda}[D_{KL}(\rho||\pi) + \log\frac{1}{\tau} + \Psi_{x_t,\pi}(\lambda, n)], \quad (16)$$

where $\Psi_{x_t,\pi}(\lambda, n) = \log \mathbb{E}_{\pi(H)}\mathbb{E}_{x_t \sim D_t}[\exp(\lambda(\mathcal{L}(H) - \tilde{\mathcal{L}}(H)))]$.

**Lemma 1** [59]. The PAC-Bayes bound, involving constants $\tau$ and $n$, as introduced in Theorem 1, is minimized by the Bayesian posterior $\rho(H)$, which represents the distribution over $\Theta$.

**Proof**. The Donsker-Varadhan's change of measure states that for any measurable function $\phi : \Theta \to \mathbb{R}$, we have:

$$\mathbb{E}_{\rho(H)}[\phi(H)] \leq D_{KL}(\rho||\pi) + \log \mathbb{E}_{\pi(H)}[\exp(\phi(H))]. \tag{17}$$

Thus, with $\phi(H) := \lambda(\mathcal{L}(H - \widetilde{\mathcal{L}}(H, x_t))$ and $\forall \rho$ over hypothesis space $\Theta$, we have:

$$\mathbb{E}_{\rho(H)}\left[\lambda\left(\mathcal{L}(H) - \widetilde{\mathcal{L}}(H, x_t)\right)\right] = \lambda\left(\mathbb{E}_{\rho(H)}[\mathcal{L}(H)] - \mathbb{E}_{\rho(H)}[\widetilde{\mathcal{L}}(H, x_t)]\right)$$

$$\leq D_{KL}(\rho||\pi) + \log \mathbb{E}_{\pi(H)}\left[\exp\left(\lambda\left(\mathcal{L}(H) - \widetilde{\mathcal{L}}(H, x_t)\right)\right)\right]. \tag{18}$$

For the non-negative random variable $\zeta_\pi(x_t) := \mathbb{E}_{\pi(H)}[\exp(\lambda(\mathcal{L}(H) - \widetilde{\mathcal{L}}(H, x_t)))]$, we apply Markov's inequality on it, and have:

$$\mathbb{P}\left(\zeta \leq \frac{1}{\tau}\mathbb{E}_{x_t \sim D_t}[\zeta_\pi(x_t)]\right) \geq 1 - \tau. \tag{19}$$

This implies that with probability at least $1 - \tau$ over the choice of $x_t \sim D_t$, we have $\forall \rho$ over hypothesis space $\Theta$:

$$\mathbb{P}\left(\mathbb{E}_{\rho(H)}[\mathcal{L}(H)] \leq \mathbb{E}_{\rho(H)}[\tilde{\mathcal{L}}_{x_t}(H)] + \frac{1}{\lambda[D_{KL}(\rho||\pi) + \log\frac{1}{\tau} + \Psi_{x_t,\pi}(\lambda, n)]}\right) \geq 1 - \tau, \tag{20}$$

where $\Psi_{x_t,\pi}(\lambda, n) = \log \mathbb{E}_{\pi(H)}\mathbb{E}_{x_t \sim D_t}[\exp(\lambda(\mathcal{L}(H) - \widetilde{\mathcal{L}}(H)))]$, and we prove the statement of Theorem 1. During target model training, as just as in Eq. (2), we utilize $\mathcal{M}_\theta$ as the prediction of posterior distribution and $S(x_i)$ as the prediction of prior distribution. Therefore, the upper bound of our target model can be expressed as:

$$\frac{1}{N_t}\sum_{i=1}^{N_t}[\mathcal{L}_{other} + \frac{1}{\lambda}D_{KL}(\mathcal{M}_\theta(x_i)||S(x_i))], \tag{21}$$

where $N_t$ is defined as the number of the target data $x_t$. Following previous BUDA works [40, 15–19], as presented in Eq. (2), $\lambda$ is set to 1 in the BUDA task. Moreover, $\mathcal{L}_{other}$ varies in different works. For example, in RrED, $\mathcal{L}_{other} = \mathcal{L}_{GC} + \mathcal{L}_{CC}$ in the first stage DTR, and $\mathcal{L}_{other} = \mathcal{L}_{GC} + \mathcal{L}_{SVI}$ in the second stage SRM. In summary, this proof fills the theoretical knowledge gap regarding the black-box predictor in previous BUDA works.

**Generalization Bound.** Since our target model is trained in unlabeled target domain data and further generates fusion data with feature differences based on the diffusion and our FSG module, we denote $x_t \sim D_t$ as the real sample distribution of the target domain and $\widetilde{x}_t \sim \widetilde{D}_t$ as the generated fusion sample distribution of the target domain. And denote $y_t$ as the predicted labels of $x_t$. For the corresponding generated fusion samples, $\widetilde{y}_t$ are the predicted labels of the target domain. $D_t$ is uploaded to a black-box predictor to obtain hard predictions from a source model trained on the source domain $D_s$, where $x_s \sim D_s$ as the sample distribution of the source domain. The pioneering study [61] on theoretical analysis for domain adaptation provide the generalization bound. Following [61], let $H$ denote a hypothesis, which can be expressed as:

$$\epsilon_t(H, y_t) \leq \epsilon_s(H, y_s) + d_{n\Delta n}(D_t, D_s) + \varphi, \tag{22}$$

where $\varphi$ denotes the shared error of the ideal joint hypothesis, $\varphi = min(\epsilon_s(H, y_s), \epsilon_t(H, y_t))$. $d_{n\Delta n}(D_s, D_t) = 2\sup_{H,H' \in n}\left|\mathbb{E}_{x_s \sim D_s}[H(x_s) \neq H'(x_s)] - \mathbb{E}_{x_t \sim D_t}[H(x_t) \neq H'(x_t)]\right|$. $\epsilon_t(H, y_t)$ is the expected error of the target sample distribution; $\epsilon_s(H, y_s)$ is the expected error of the source sample distribution, which is obtained from the black-box predictor. In the BUDA setting, although we do not have the source domain data $x_s$, we can obtain hard predictions $P_s$ from the black-box predictor. Therefore, according to the theory [62, 63] of source data absence, $\epsilon_s(H, y_s)$ is small and can be ignored, so we do not need to obtain $x_s$ and $y_s$ in BUDA.

Then, we model a generated fusion domain distribution $\widetilde{D}_t$ that is distributed similarly to the target distribution $D_t$. To reduce the classification error on the target domain, the distributions $D_s$, $D_t$, and $\widetilde{D}_t$ should be substantially similar to each other. Therefore, the generalization bound in RrED can be transformed into:

$$\epsilon_t(H, y_t) \leq \widetilde{\epsilon}_t(H, \widetilde{y}_t) + d_{n\Delta n}(D_t, \widetilde{D}_t) + \varphi_1, \tag{23}$$

where $\varphi_1 = min(\epsilon_t(H, y_t), \widetilde{\epsilon}_t(H, \widetilde{y}_t))$; $\widetilde{\epsilon}_t(H, \widetilde{y}_t)$ is the expected error of the generated fusion domain distribution, which can be expressed as:

$$\widetilde{\epsilon}_t(H, \widetilde{y}_t) \leq \epsilon_s(H, y_s) + d_{n\Delta n}(\widetilde{D}_t, D_s) + \varphi_2, \tag{24}$$

where $\varphi_2 = min(\widetilde{\epsilon}_t(H, \widetilde{y}_t), \epsilon_s(H, y_s))$. Thus, our final generalization bound can be defined as:

$$\epsilon_t(H, y_t) \leq \epsilon_s(H, y_s) + d_{n\Delta n}(D_t, \widetilde{D}_t) + d_{n\Delta n}(\widetilde{D}_t, D_s) + \varphi_1 + \varphi_2. \tag{25}$$

For Eq. (25), we analyze each component in detail in this paragraph:

• $\epsilon_s(H, y_s)$ is the expected error of the source sample distribution. During the training of the source model, the error between the source domain data and its true labels is minimized by cross-entropy loss. Thus, in the early stages of training, we can obtain good training results for the source samples through the black-box predictor. As the training progresses, the model gradually adapts to the distribution of the target domain with Adaptive Label Smoothing (ALS) [15]. The ALS maintains source domain knowledge, enabling the model to learn target domain knowledge while preventing the forgetting of source domain knowledge. Therefore, according to theories [62, 63], $\epsilon_s(H, y_s)$ is small in the whole training.

• Instead of reducing $d_{n\Delta n}(D_t, D_s)$ in Eq. (16), our goal is to reduce $d_{n\Delta n}(D_t, \widetilde{D}_t)$ and $d_{n\Delta n}(\widetilde{D}_t, D_s)$. For $d_{n\Delta n}(D_t, \widetilde{D}_t)$, it depends on the expected error of the disagreement between two hypothesis on the target data and the generated fusion data distribution of the target domain. During the whole training, we design the FSG module to determine which regions should be composed of synthetic images. $\widetilde{D}_t$ is generated from $D_t$, preserving key features of $D_t$ while adding differential features generated by the diffusion model. Therefore, the distribution divergence $d_{n\Delta n}(D_t, \widetilde{D}_t)$ is small. For $d_{n\Delta n}(\widetilde{D}_t, D_s)$, we can obtain that $d_{n\Delta n}(\widetilde{D}_t, D_s) = 2\sup_{H, H' \in n} \left| \mathbb{E}_{\widetilde{x}_t \sim \widetilde{D}_t}\left[H(\widetilde{x}_t) \neq H'(\widetilde{x}_t)\right] - \mathbb{E}_{x_s \sim D_s}\left[H(x_s) \neq H'(x_s)\right] \right|$. As the training progresses, DTR aligns $x_t$ and $\widetilde{x}_t$ by continuously minimizing the cross-entropy loss to facilitate the target model's training; SRM narrows the feature space distance between $x_t$ and $\widetilde{x}_t$ by contrasting their differences, while enhancing the model's discriminative and generalization abilities by increasing dissimilarities with other samples. Therefore, $\mathbb{E}_{\widetilde{x}_t \sim \widetilde{D}_t}\left[H(\widetilde{x}_t) \neq H'(\widetilde{x}_t)\right] \approx \mathbb{E}_{x_t \sim D_t}\left[H(x_t) \neq H'(x_t)\right]$ and it is continuously reduced during training by minimizing $\mathcal{L}_{GC}$ and $\mathcal{L}_{task}$. Meanwhile, $\mathcal{L}_{CC}$ and $\mathcal{L}_{SVI}$ prevent overfitting of the target model. For $\mathbb{E}_{x_s \sim D_s}\left[H(x_s) \neq H'(x_s)\right]$, according to the previous works [15, 16], the ALS maintains a source knowledge base and use $\mathcal{L}_{task}$ to maintain the balance between source knowledge and target knowledge. Therefore, $\mathbb{E}_{x_s \sim D_s}\left[H(x_s) \neq H'(x_s)\right]$ always maintains a small value during the whole adaptation phase.

• $\varphi_1 + \varphi_2$ denotes the shared error of the ideal joint hypothesis, which is assumed to be a sufficiently small constant that reflects the complexity of the hypothesis space [62].

## D. The Whole Training Process

Our pseudocode for the training process is shown in Algorithm 1. In addition, our experimental and main code are available in the Supplementary Material.

## E. Supplement of Complete Quantitative Ablation Experimental Results

As shown in Table 6, we report the complete quantitative results of ablation, and all the results include the task-specific loss. FSG is the key module of our work to prevent the diffusion-generated images from causing irreversible negative effects. When FSG and FT are not used, the results indicate that directly applying the default Stable Diffusion model, without adaptation to the downstream task, leads to a sharp performance drop. In contrast, our method exhibits highly task-aware sensitivity to the structural characteristics of the Stable Diffusion model, enabling it to better leverage its semantic knowledge for downstream BUDA tasks. The effective knowledge learning of the target model through $\mathcal{L}_{GC}$ and $\mathcal{L}_{CC}$ can only be achieved when FSG is utilized. When $\mathcal{L}_{CC}$ is not used, the combined effect of $\mathcal{L}_{task}$ and $\mathcal{L}_{GC}$ enforces rapid sample clustering, which leads to overfitting of the target model. $\mathcal{L}_{CC}$ mitigates the sample enrichment effect to improve target model's generalization. In this regard, both $\mathcal{L}_{task}$ and $\mathcal{L}_{GC}$ can benefit from this process. $\mathcal{L}_{SVI}$ is to integrate interactive

**Algorithm 1** RrED for BUDA task.

---

**Input:** Target samples $D_t = \{(x_i)\}_{i=1}^{N_t}$; black-box hard predictions $P_s$; diffusion model with the predictor $p_\theta$; multi-modal model $\mathcal{V}_\theta \in$ {image encoder $\mathcal{I}_\theta$, text encoder $\mathcal{T}_\theta$}; and target model $\mathcal{M}_\theta \in$ {feature extractor $f_\theta$, prediction classifier $c_\theta$}.

**Parameter:** Training epoch $e$; learnable prompt text embedding $L$; model parameter $\theta$; and hyperparameters $\gamma$, $r$.

1: **Initialize:** initialize the smooth label repository $S$ with $P_s$; initialize $\mathcal{V}_\theta$ with $L$ and $S$; diffusion model initializes to generate data $x_{i,(g)}$ corresponding to $x_i$;
2: ===================== **Diffusion-Target model Rectification** =====================
3: **for** $i \leftarrow 1$ **to** $e/2$ **do**
4:     Get target sample $x_i$ and the sample predictions $y_i$ using $\mathcal{M}_\theta$: $y_i = f_\theta(c_\theta(x_i))$;
5:     Get generated fusion sample $\widetilde{x}_i$ to fuse $x_{i,(g)}$ and $x_i$ using Eqs. (3)-(4);
6:     Update the smooth label repository $S$ using Eq. (1);
7:     Get fusion sample predictions $\widetilde{y}_i$ using $\mathcal{M}_\theta$: $\widetilde{y}_i = f_\theta(c_\theta(\widetilde{x}_i))$;
8:     Fine-tune $\mathcal{V}_\theta$ by minimizing $\mathcal{L}_{\mathcal{V}_\theta}$ with $p_\theta(x_i)$ using Eqs. (5)-(6): $\min_{\mathcal{I}_\theta} \max_{\mathcal{T}_\theta} \mathcal{L}_{\mathcal{V}_\theta}$;
9:     Optimize $\mathcal{M}_\theta$ by minimizing $\mathcal{L}_{\mathcal{M}_\theta(DTR)}$ with $p_\theta(x_i)$ using Eq. (9): $\min_{f_\theta} \max_{c_\theta} \mathcal{L}_{\mathcal{M}_\theta(DTR)}$;
10: **end for**
11: ===================== **Self-Rectifying Reasoning Model** =====================
12: **Initialize:** Replace the original text encoder in the diffusion model with the fine-tuned text encoder with prompt word embeddings;
13: **for** $i \leftarrow e/2$ **to** $e$ **do**
14:     Get target sample $x_i$ and the sample predictions $y_i$ using $\mathcal{M}_\theta$: $y_i = f_\theta(c_\theta(x_i))$;
15:     Get generated fusion sample $\widetilde{x}_i$ to fuse $x_{i,(g)}$ and $x_i$ using Eqs. (3)-(4);
16:     Update the smooth label repository $S$ using Eq. (1);
17:     Get fusion sample predictions $\widetilde{y}_i$ using $\mathcal{M}_\theta$: $\widetilde{y}_i = f_\theta(c_\theta(\widetilde{x}_i))$;
18:     Assign different weights $w$ according to the similarities between $y_i$ and $\widetilde{y}_i$ using Eq. (11);
19:     Optimize $\mathcal{M}_\theta$ by minimizing $\mathcal{L}_{\mathcal{M}_\theta(SRM)}$ with $w$ using Eq. (12): $\min_{f_\theta} \max_{c_\theta} \mathcal{L}_{\mathcal{M}_\theta(SRM)}$;
20: **end for**
**Output:** Target model $\mathcal{M}_\theta$.

Table 6: The complete quantitative results of ablation study on the *Office-31* and *VisDA-17*.

| $\mathcal{L}_{GC}$ | $\mathcal{L}_{CC}$ | $\mathcal{L}_{SVI}$ | FSG | FT | Office-31 A→D | A→W | D→A | D→W | W→A | W→D | Mean | VisDA-17 Mean |
|---|---|---|---|---|---|---|---|---|---|---|---|---|
| | Source only | | | | 79.9 | 76.6 | 56.4 | 92.8 | 60.9 | 98.5 | 77.5 | 48.9 |
| ✓ | | | | ✓ | 97.8 | 85.2 | 66.6 | 97.0 | 72.1 | 97.5 | 86.0 | 71.2 |
| | ✓ | | | ✓ | 85.5 | 94.9 | 79.3 | 99.1 | 83.5 | 99.8 | 90.4 | 79.3 |
| ✓ | ✓ | | | | 76.2 | 83.2 | 67.6 | 94.1 | 69.2 | 95.6 | 81.0 | 59.6 |
| ✓ | ✓ | | | ✓ | 95.2 | 95.7 | 81.5 | 99.0 | 83.1 | 99.8 | 92.4 | 89.4 |
| | | ✓ | | ✓ | 92.7 | 88.5 | 67.7 | 97.9 | 74.5 | 99.6 | 86.9 | 67.8 |
| ✓ | | ✓ | | ✓ | 96.9 | 94.1 | 73.7 | 97.3 | 81.6 | 99.8 | 90.6 | 85.4 |
| | ✓ | ✓ | | ✓ | 85.3 | 84.9 | 66.7 | 97.0 | 71.9 | 98.0 | 84.0 | 80.2 |
| | ✓ | ✓ | ✓ | ✓ | 87.3 | 84.0 | 65.6 | 96.3 | 74.2 | 98.6 | 84.3 | 81.7 |
| ✓ | ✓ | ✓ | | | 71.7 | 84.9 | 70.2 | 94.6 | 64.4 | 98.2 | 80.6 | 75.9 |
| ✓ | ✓ | ✓ | ✓ | | 96.8 | 94.1 | 82.5 | 99.1 | 84.1 | 99.8 | 92.7 | 88.7 |
| ✓ | ✓ | ✓ | | ✓ | 73.1 | 85.7 | 69.7 | 95.3 | 65.2 | 97.9 | 81.2 | 76.8 |
| ✓ | ✓ | ✓ | ✓ | ✓ | **97.8** | **95.9** | **83.7** | **99.1** | **84.5** | **99.8** | **93.5** | **91.2** |

learning with the samples generated by the fine-tuned diffusion model. $\mathcal{L}_{SVI}$ becomes effective only when combined with fine-tuning. Experimental results show that this combination yields significant performance gains on large-scale dataset *VisDA-17*, while improvements on small-scale dataset *Office-31* are relatively limited. In summary, each component of our RrED contributes effectively to performance improvement and is indispensable.

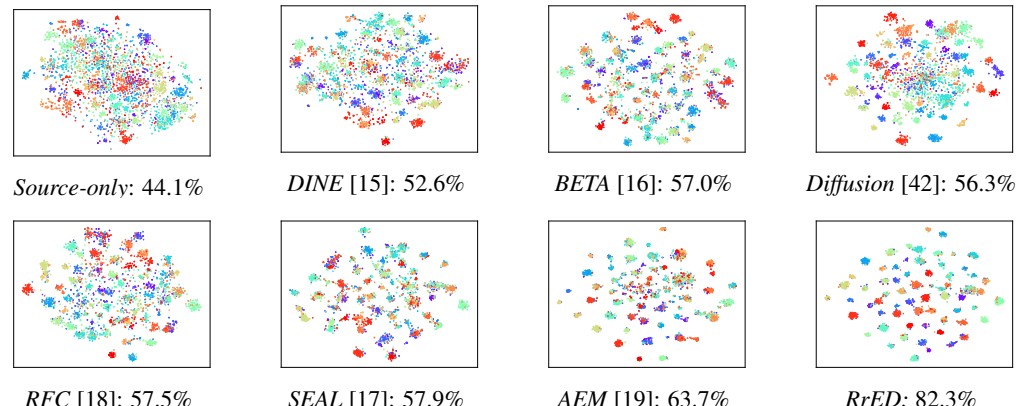

Figure 6: The feature visualization on the Office-Home (A→C) using the t-SNE [65]. Herein, the points represent target samples and the different colors correspond to their ground-truth classes. RrED introduces diffusion into BUDA and fine-tunes it, ultimately achieving remarkable improvement.

## F. Implementation Details.

We implement our RrED based on PyTorch and conduct all experiments using an NVIDIA GeForce RTX4090 GPU. For fair comparison, the backbone network is initialized following the protocol in [15], employing the ImageNet [64] pre-trained ResNet architectures: ResNet-50 for *Office-31*, *Office-Home*, and *DomainNet*, and ResNet-101 for *VisDA-17*. The optimization configuration employs SGD with a momentum of 0.9, a weight decay of 1e-3, and differentiated learning rates, where the learning rate is set to 1e-4 for the feature extractor $f_\theta$ and 1e-3 for the classifier $c_\theta$. Following [16, 17], we set the bottleneck dimension to 256, the batch size to 64, the static momentum coefficient $\mu$ to 0.6, and the number of warm-up epochs to 3. To facilitate our joint multi-modal model CLIP [20] for fine-tuning, we choose Stable Diffusion v-1.5 [22] as the diffusion model. The strength of the noise addition is set to 0.6 in the diffusion model. For the diffusion predictor [42] and the fine-tuned text encoder we introduced, we keep their parameters frozen during the whole training. During the fine-tuning process, we follow [52, 19] to set the number of context tokens $m$ to 4. All the reported quantitative results are obtained by averaging multiple runs with seeds [2023, 2024, 2025].

## G. More Visual Comparisons and Further Analysis

As shown in Figure 6, we use t-SNE [65] technique to visualize the distribution of target samples in the feature space. Compared with previous methods, the discrimination ability of the target model for target samples with similar features has been significantly improved under the training of our RrED algorithm. Moreover, as can be clearly observed from the graph, due to the enhanced generalization ability of the model after being trained by RrED, the differences between different classes become more pronounced, and the distances between samples of the same class become more compact. Compared to the previous method [42] that directly applies diffusion model for prediction, our RrED exhibits superior model generalization and class discrimination capabilities. Therefore, we conclude that the target model trained by RrED achieves significant performance improvement in the high-security BUDA setting.

Next, we discuss our method's exploration of the diffusion model to further demonstrate the superiority of our approach. As shown in Figure 7, we show the images that are generated by the diffusion model on the *VisDA-17* under varying noise strengths. When the noise level is too low, the images generated by the diffusion model are too similar to the target domain images, providing limited benefit for enhancing the model's reasoning ability. When the noise level is too high, the images generated by the diffusion model differ drastically from those in the target domain and may even contain unrelated objects. Directly using such images can irreversibly disrupt the discriminative ability of the target model. *How can we effectively utilize the diffusion model to guide the target model in enhancing its reasoning ability while preventing its potential negative effects?* This is the problem our work RrED aims to solve. For this, FSG serves as the key module to preclude the diffusion-generated

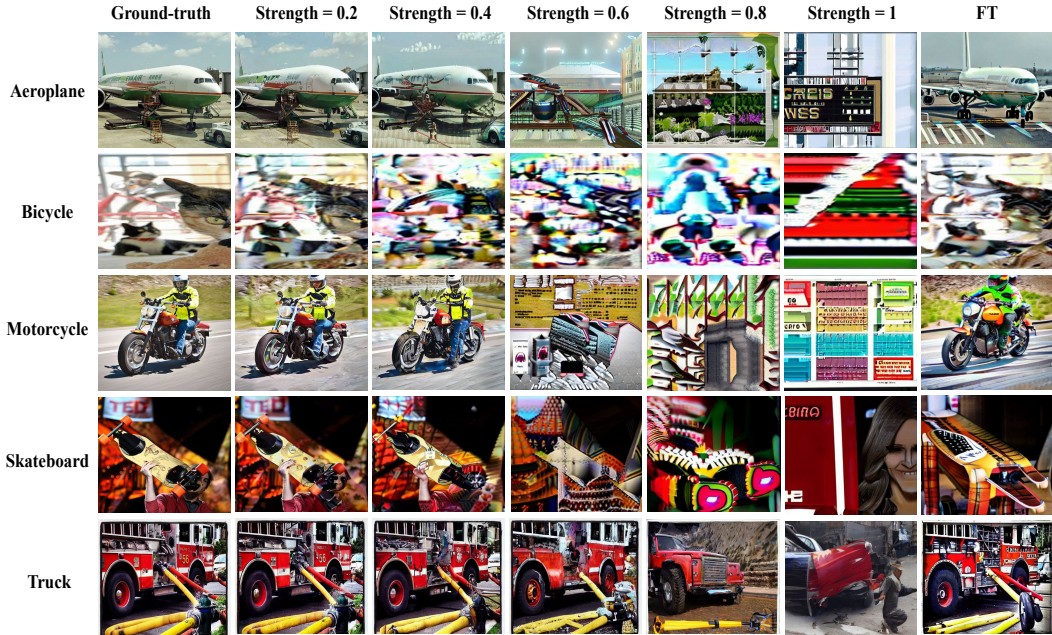

|  | Ground-truth | Strength = 0.2 | Strength = 0.4 | Strength = 0.6 | Strength = 0.8 | Strength = 1 | FT |

Figure 7: We present images generated by the diffusion model on the *VisDA-17* under varying noise strengths, along with those produced when the noise strength is 0.6 after our fine-tuning.

Table 7: Results of computational cost comparison on the *VisDA-17* with the ResNet-101 backbone. The batch size is 64.

| Method | Space (MiB) | Time (s/epoch) | Accuracy (%) |
|---|---|---|---|
| DINE | 9881MiB | 124s | 75.6 |
| BETA | 20247MiB | 1101s | 85.1 |
| SEAL | Over 24G | - | 89.2 |
| AEM | 13747MiB | 672s | 89.3 |
| RrED (Stage 1) | 17654MiB | 312s | 89.4 |
| RrED (Stage 2) | 11721MiB | 201s | 91.2 |

images from bringing about irreversible adverse effects. FSG retains the regions of interest for the model, allowing the target model to maintain image discernibility even under higher noise levels in diffusion. Meanwhile, by fine-tuning the text encoder in the diffusion model, RrED enables it to better understand the content to be generated while maintaining its generative capabilities. As shown in Figure 7, the images generated by the fine-tuned diffusion model exhibit greater diversity, more distinct features, and fewer interfering objects. This allows the target model, in the second phase SRM, to first recognize the simpler generated images and then further distinguish the more challenging target images.

## H. Computational Cost Comparison and Optimization Evolution

We supplement the computational cost comparison of the *VisDA-17* [55] in Table 7. For a fair comparison, all the results are obtained in the same experimental environment. In Table 7, we document the maximum GPU space usage, the average runtime cost, and the best accuracy of each comparison method. When adapting to the *VisDA-17* dataset, it is worth noting that the comparison methods have consumption-related limitations. BETA [16] operates in two computationally intensive stages: the first stage is the initialization, which requires initialization of the two models due to their mutually-distilled network structures; the second stage is the two-step process, which requires distillation and fine-tuning for each epoch. SEAL [17] is highly resource-intensive, and its official code cannot complete the adaptation task on *VisDA-17* under the same conditions with 24GB GPU memory. During the training of AEM [19], two classifiers are required: one classifier processes the output of the target model, while the other aligns with the predictions of the ViL model. Moreover, in

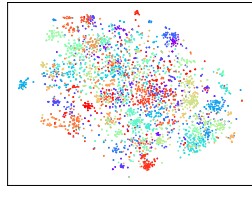 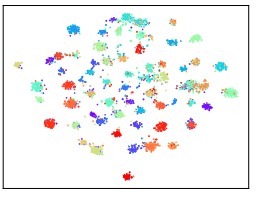 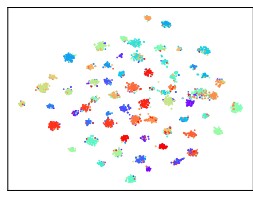

*Source-only*: 44.1%          *Stage 1*: 79.8%          *Stage 2*: 82.3%

Figure 8: The feature distribution evolutions of different stages on the *Office-Home* (A→C) using t-SNE [65]. Herein, the points represent target samples and the different colors correspond to their ground-truth classes.

each iteration, the weights of the overall model and the classifier weights need to be updated separately, resulting in consuming a significant amount of time. Compared with the previous BUDA methods, although RrED introduced the diffusion model in stage 1 to guide the learning of the target model, it still significantly reduced the time consumption by cutting out unnecessary calculation processes and optimizing loss functions. Moreover, in stage 2, after eliminating the resources consumed by fine-tuning and diffusion, RrED demonstrates extremely low overhead. These results demonstrate that our RrED significantly outperforms other BUDA methods in enhancing class discrimination ability at a relatively low cost.

In Figure 8, the optimization evolutions of feature distribution are presented. The black-box predictor fails to effectively separate and cluster target sample features, with samples from different classes heavily entangled. This confusion introduces noisy signals during target model training, thereby hindering effective adaptation. After the first stage of training, the target model has learned the rich semantic knowledge in the diffusion model and significantly improved its class discrimination ability. After the second stage of training, the scattered data distribution boundaries stabilize around the nearest feature cluster centers, thus leading to the samples with similar features exhibiting a more compact behavior. These results demonstrate the superiority of the two-stage training in RrED and achieve the predefined objectives of each stage.

## I. Broader Impacts and Limitations

Our work RrED focuses on the problem of Black-box Unsupervised Domain Adaptation (BUDA), which provides better data privacy protection with more flexible portability compared with other Domain Adaptation (DA) settings. Meanwhile, RrED demonstrates extremely superior performance, significantly surpassing other DA methods. Inspired by research in neuroscience, RrED is specifically designed for the classification task. While its effectiveness has been demonstrated through extensive experiments and its theoretical soundness established, its applicability to other tasks remains an open question. Therefore, we plan to further explore the practical utility of this algorithm in a broader range of task scenarios.

