# OpenReview forum: "RrED: Black-box Unsupervised Domain Adaptation via Rectifying-reasoning Errors of Diffusion"
_NeurIPS.cc/2025/Conference — NeurIPS 2025 poster_

### Official Review · Reviewer_H3mL · 2025-06-19

**Clarity:** 2
**Significance:** 2
**Originality:** 3
**Rating:** 4
**Confidence:** 4

**Summary:**

This paper addresses the Black-box Unsupervised Domain Adaptation (BUDA) problem by introducing a novel diffusion-based framework, RrED (Rectifying-reasoning Errors of Diffusion). The method is structured in two stages: (1) Diffusion-Target model Rectification (DTR) and (2) Self-rectifying Reasoning Model (SRM). The first stage decouples and fine-tunes the diffusion model’s text encoder with a feature-sensitive generation (FSG) module, while the second stage uses the model’s prediction discrepancy across generated variations to refine its reasoning ability. Extensive experiments across four standard domain adaptation benchmarks show that RrED outperforms state-of-the-art (SOTA) BUDA methods.

**Questions:**

Please see the weakness above. Overall, this work presents a technically complex framework for black-box unsupervised domain adaptation, but its main weakness lies in the lack of clear motivation. The paper does not sufficiently justify the choice of diffusion models or explain the conceptual rationale behind the proposed design. In addition, the presentation should be improved to clearly convey what specific problem is being addressed in BUDA and how the proposed method aims to solve it.

**Ethical Concerns:**

["NO or VERY MINOR ethics concerns only"]

**Final Justification:**

The authors’ response has addressed my concern, and I also hope the authors will take the reviewers’ comments into account during the revision. After considering the other reviewers’ comments, I will give a borderline accept.

**Limitations:**

yes

**Quality:**

3

**Strengths And Weaknesses:**

**Strength**

1. Employing diffusion model for tackling BUDA task is novel.

2. The authors perform thorough evaluations on four well-established datasets (Office-31, Office-Home, VisDA-17, DomainNet), showing consistent improvements.

3. Comprehensive ablation studies are conducted to validate the contributions of each component. The inclusion of qualitative visualizations (e.g., Grad-CAM) is helpful.

**Weakness**

1. This work presents a complex framework but lacks important and necessary analysis of the underlying intuition:

(1) Insufficient Justification for Diffusion Models in BUDA: The paper lacks a detailed explanation of the motivation for using diffusion models to address the black-box unsupervised domain adaptation (BUDA) task. Specifically, it does not clearly explain why diffusion models are suitable for BUDA, what particular challenges in BUDA they aim to address, and in what aspects diffusion models are advantageous compared to other multimodal models such as CLIP.

(2) Weak Theoretical Connection to Human Reasoning: The paper claims that the proposed method is inspired by the human decision-making process, specifically the dual-process theory involving System 1 and System 2. However, the connection between the method and this cognitive framework remains vague. The authors should provide more discussion on how the proposed components correspond to these cognitive systems. In particular, it is unclear what the term ‘reasoning’ specifically refers to in the context of BUDA. Does it involve rectifying predicted labels for target data?


2. Lack of Qualitative Reasoning Examples: The model is claimed to enhance "reasoning" yet no explicit qualitative examples or visualizations are provided to show what these reasoning steps are. Without this, the claim that the model improves reasoning over past BUDA methods is difficult to assess.

---

> ### Author Rebuttal · Authors · 2025-07-30
>
> We sincerely appreciate the reviewer’s constructive suggestions and have answered them as follows with our highest respect.
>
> Q1: Insufficient Justification for Diffusion Models in BUDA: The paper lacks a detailed explanation of the motivation for using diffusion models to address the black-box unsupervised domain adaptation (BUDA) task. Specifically, it does not clearly explain why diffusion models are suitable for BUDA, what particular challenges in BUDA they aim to address, and in what aspects diffusion models are advantageous compared to other multimodal models such as CLIP.
>
> A1: We have provided a detailed explanation of the motivation for using diffusion models to address the BUDA task. In Table 1 and lines 65-67, our motivation is to address the limitations of current diffusion-based UDA methods in terms of privacy protection and computational overhead. While diffusion models contain rich semantic knowledge that can guide BUDA learning, prior diffusion-UDA approaches focus solely on the image encoder, neglecting the role of the text encoder, thus failing to provide effective supervision for BUDA. Please refer to lines 78-91, our RrED is the first work that applies the diffusion model to BUDA tasks, which is a pioneering attempt. Please refer to lines 691-708, we have explained the limitations of directly applying diffusion models in BUDA tasks, and further clarified why our proposed method can achieve effective adaptation and performance improvement under this setting. Compared to other multimodal models, diffusion models offer the unique advantage of generating stable images. In addition, we qualitatively illustrate the differences between our approach and the CLIP-based method AEM [1] in Figure 3. In 562-567, We have provided a detailed explanation, from a class-level perspective, of why our diffusion model-based method holds advantages over other multimodal approaches. Furthermore, in lines 719–728, we have provided a detailed quantitative analysis demonstrating our advantages over AEM. Therefore, we have clearly explained the motivation for employing diffusion models in the BUDA setting and provided sufficient experimental evidence to validate its effectiveness and rationality.
>
> Q2:  Weak Theoretical Connection to Human Reasoning: The paper claims that the proposed method is inspired by the human decision-making process, specifically the dual-process theory involving System 1 and System 2. However, the connection between the method and this cognitive framework remains vague. The authors should provide more discussion on how the proposed components correspond to these cognitive systems. In particular, it is unclear what the term ‘reasoning’ specifically refers to in the context of BUDA. Does it involve rectifying predicted labels for target data?
>
> A2: As shown in 68-77 and Figure 1, we have provided a detailed description of the connection between our method and the human decision-making system. To address the reviewers' concerns, we provide more discussion on how our method corresponds to the cognitive system. For the term “reasoning”, in human decision-making systems, reasoning ability refers to the capacity of individuals to make further judgments about target objects by leveraging prior knowledge and logical analysis based on partial observations when confronted with complex information and dynamic environments. Similarly, in computer vision, the model’s reasoning process mirrors human decision-making by utilizing existing knowledge and feature similarity computations to determine whether targets in complex scenes meet the task requirements. The reasoning-related knowledge is relatively common. In neuroscience, human decision-making involves two stages: Stage 1 generates intuitive but sometimes biased responses unconsciously, while Stage 2 uses domain knowledge for careful, accurate reasoning. Recent studies add knowledge bases to guide this process and reduce reasoning errors. Our RrED models this by first aligning diffusion model decision boundaries in Stage 1, then correcting errors between versions in Stage 2 to improve self-reasoning. Meanwhile, inspired by previous approaches [3, 4] that enhance decision-making and reasoning, RrED introduce the diffusion model as a knowledge base. By leveraging its rich semantic representations, we guide the learning of the target model to identify and correct potential errors in its predictions. To improve the clarity and readability of the paper, we plan to supplement this knowledge in the Appendix.
>
> Q3: Lack of Qualitative Reasoning Examples: The model is claimed to enhance "reasoning" yet no explicit qualitative examples or visualizations are provided to show what these reasoning steps are. Without this, the claim that the model improves reasoning over past BUDA methods is difficult to assess.
>
> A3: In the field of computer vision, GAM-Grad [2] is a well-known technique for interpreting the reasoning behind model predictions through visualization. Leveraging its capabilities, we employ GAM-Grad in Figure 3 to highlight the vital and irreplaceable roles each component of our algorithm plays in the overall performance, with detailed explanations provided in lines 318-323. Furthermore, in Figure 3, we compare our approach with the previous SOTA method AEM, demonstrating that our model exhibits stronger reasoning ability, superior target discrimination, and more precise capture of feature details in target samples. To help reviewers more intuitively understand how our method improves the model’s reasoning ability during execution, we plan to include the initial and post-stage 1 Grad-CAM visualizations in the final version.
>
> We appreciate the reviewer’s feedback and hope our response has addressed the reviewer’s concerns.
>
> [1] Adversarial Experts Model for Black-box Domain Adaptation. In ACMMM, 2024.
>
> [2] Grad-cam: Visual explanations from deep networks via gradient-based localization. In CVPR, 2017.
>
> [3] Efficient rectification of neuro-symbolic reasoning inconsistencies by abductive reflection. In AAAI, 2025.
>
> [4] Abductive learning with ground knowledge base, In IJCAI, 2021.

---

> > ### Comment · Reviewer_H3mL · 2025-08-03
> >
> > Thanks for the response. After reading the rebuttal, I have a few additional suggestions:
> >
> > For Q1, I suggest that the authors explicitly discuss the motivation for using diffusion models for BUDA earlier in the paper. This would help readers better understand the relevance and importance of diffusion models to the BUDA problem, rather than focusing solely on how existing diffusion-based UDA methods do not directly apply to the BUDA setup. The current framing may weaken the importance of diffusion models for BUDA.
> >
> > For Q2, my concern is that most existing works that emphasize "reasoning" are based on large language models (e.g., improving the reasoning abilities of LLMs to think more like humans), which could be confusing for readers. I recommend making an early clarification or claim to distinguish between these different types of reasoning.

---

> ### Author Response · Authors · 2025-08-04
> **Thank the reviewer for the valuable suggestion.**
>
> We sincerely thank the reviewer for the valuable suggestion.
>
> In response to suggestion 1, we add more detailed motivation in the Introduction section to declare the significance of the diffusion model for the BUDA task: “Compared to other multimodal models, diffusion models not only have rich semantic knowledge but also can stably generate diverse images. In the BUDA setting, such stable generation helps enhance model generalization by providing consistent image-level augmentation. **How to effectively leverage diffusion models in BUDA tasks to guide the target model in enhancing its reasoning ability while preventing its potential negative effects?**  This is one of our research motivations”.
>
> In response to suggestion 2, we add more detailed explanations in the Recent Works section to emphasize that "reasoning" is specific to the field of computer vision: “The reasoning ability of LLMs refers to their capacity for abstract and logical inference, whereas in computer vision, the reasoning ability focuses on identifying input regions that most influence the model’s decision to infer the most probable target class”.
>
> Sincerely, we hope our response has addressed all concerns and wins the positive score of the reviewer.

---

> ### Author Response · Authors · 2025-08-05
> **Looking forward to your feedback.**
>
> Dear reviewer H3mL,
>
> We hope this message finds you well. We have improved our paper based on your constructive suggestions. As the discussion period is nearing its end, we would like to ensure that all your concerns have been addressed satisfactorily. If there are any additional points or feedback, please let us know. Your insights are invaluable to us, and we’re eager to address any potential issues to improve our work.
>
> Thank you for your time and effort in reviewing our paper.
>
> Sincerely,
>
> Authors

---

### Official Review · Reviewer_mM7d · 2025-06-22

**Clarity:** 1
**Significance:** 3
**Originality:** 2
**Rating:** 4
**Confidence:** 3

**Summary:**

This paper aims to adapt a model in the setting of Black-box Unsupervised Domain Adaptation (BUDA), which doesn't allow to access source data and source model. It proposes a two-stage framework leveraging diffusion models as a "knowledge base" to enhance target model adaptation. In the first stage, it designed feature sensitive generation to fuse new samples and finetune the text encoder of diffusion model to generate more stable target samples. Then in the second stage, it contrasts predictions of original vs. diffusion-augmented samples to refine decision boundaries. Overall, the framework adapt a model achieve better classification results in several benchmarks.

**Questions:**

Just like items said in Weaknesses:
1. How to effectively manage so many modules to make it stable in actual scenarios?
2. Supplement analysis and intuitive examples for the immediate and innovative designs.

**Ethical Concerns:**

["NO or VERY MINOR ethics concerns only"]

**Final Justification:**

After rebuttal and further response from authors, I decided to increase my rating from 3 to 4. The main reasons are:
1. The task setting of BUDA is interesting and might have important application in widely used close-source large models.
2. After rebuttal, although it is not perfect, most of my concerns about the whole pipeline and hyperparameter selection have been resolved.
3. For the claim of connection between the method and visual reasoning, it is still not convincing. But the empirical results fundamentally verify the effectiveness.
Comprehensively, incorporating all the positive and negetive factors, I made such a decision.

**Limitations:**

yes

**Paper Formatting Concerns:**

No Paper Formatting Concerns.

**Quality:**

3

**Strengths And Weaknesses:**

Strengths:
1. Clear and important problem setting. The Black-box Unsupervised Domain Adaptation pinpoint the significance of privact preservation. Later it might be very useful in the widespread usage of closed-source large models.
2. Creative data-in-the-loop pipeline design. With the proposed fusion of samples and finetuning diffusion model, it integrates data manipulation into the adaptation process.
3. Abundant empitical and theoretical analysis. It provides classification results of multiple benchmarks and proofs for error bound.

Weaknesses:
1. The organization and description are hard to follow. There are too many abbreviations for the problem setting, the method and each module or blocks, RrED, BUDA, FSG, DTR, SVI, .etc. It is hard to build connections among them. Besides, in section 3.2, the notations are also very intricate, which leads to barrier for understanding.
2. Tanglesome design of many modules in the whole pipeline. FIgure 2 contains too many blocks, which is hard to filter key steps. Integration of diffusion model, multi-modal model, and at least 2 contrastive related losses, extremely increase the comlexity of the whole system.
3. Hyperparameter Sensitivity. As the last point said, a lot of losses in the two stages meet challenges of balance. Just like Figure 4 shows, the same hyperparameter r leads to a large difference of performance across different datasets. It hinders further application in actual scenarios.
4. Lack of quantitative and qualitative analysis of key innovative designs. After hard reading, it can find that the Feature Sensitive Generation, Finetuning diffusion models, and Sample-version Interaction are key innovative modules. Besides final classification performance, we don't know what intuitive and direct result these designs lead to. For example, what do the fused examples after Feature Sensitive Generation really look like? Although Figure 7 gives some samples generated by diffusion model, it is hard to know the difference before and after finetuning.

---

> ### Author Rebuttal · Authors · 2025-07-30
>
> We sincerely appreciate the reviewer’s constructive suggestions and have answered them as follows with our highest respect.
>
> Q1: The organization and description are hard to follow. There are too many abbreviations for the problem setting, the method and each module or blocks, RrED, BUDA, FSG, DTR, SVI, .etc. It is hard to build connections among them. Besides, in section 3.2, the notations are also very intricate, which leads to barrier for understanding.
>
> A1: Although our paper includes a number of abbreviations, each has been clearly defined and thoroughly introduced. For all equations, particularly those in Section 3.2, we have provided detailed explanations of their processes and their connections to our proposed method. Furthermore, all notations have been clearly interpreted following common practices in the machine learning community. To ensure the content is clear and easy to understand, we have carefully reviewed the content multiple times to avoid any omissions in the explanations of the final version.
>
> Q2: Tanglesome design of many modules in the whole pipeline. FIgure 2 contains too many blocks, which is hard to filter key steps. Integration of diffusion model, multi-modal model, and at least 2 contrastive related losses, extremely increase the comlexity of the whole system.
>
> A2: Although our method appears relatively complex, each step is specifically designed to address limitations inherent to the current task and is essential to the overall framework. We have modularized key components for better clarity and usability. The method can be divided into two core components: fine-tuning and integrating the diffusion model’s text encoder, and the two-stage learning pipeline for the target model. In the first stage, the diffusion model generates initial images and guides the target model using its classifier, while the text encoder is fine-tuned. In the second stage, the fine-tuned text encoder is reintegrated into the diffusion model to regenerate images for contrastive learning by the target model. We encapsulate the diffusion model operations into two functions, each corresponding to a learning stage, enabling high reusability and portability. As shown in Figure 2, every step of the process is clearly illustrated. To further improve clarity and structure, we have simplified Figure 2 by removing common operations in BUDA and optimizing the overall layout.
>
> Q3: Hyperparameter Sensitivity. As the last point said, a lot of losses in the two stages meet challenges of balance. Just like Figure 4 shows, the same hyperparameter r leads to a large difference of performance across different datasets. It hinders further application in actual scenarios.
>
> A3: For “As the last point said, a lot of losses in the two stages meet challenges of balance”, our method adopts a two-stage structure where, in addition to the task-specific loss, the first stage involves two losses and the second stage only one. In the first stage, $L_{CC}$ is a standard cross-entropy loss, while $L_{CC}$ is introduced to counteract the potential negative impact of cross-entropy minimization on generalization, with the hyperparameter $\gamma$ controlling the contribution of $L_{CC}$. In the second stage, $L_{SVI}$ is a contrastive loss that achieves self-balancing by evaluating sample similarity through the similarity weight $w$. Therefore, the loss functions introduced in each stage are not overly complex and are carefully designed to ensure training stability and balance.
>
> For “the same hyperparameter r leads to a large difference of performance across different datasets“, various factors influence performance across different datasets. In Figure 7, we compare VisDA-17 and Office-Home (A→C), where the number of target domain samples differs significantly: VisDA-17 contains around 55K samples, while Office-Home has only about 4K. As such, a large performance gap between the two is expected. In addition, the hyperparameter $r$, which determines the ratio between regions of interest and non-interest, plays a critical role in the effectiveness of our FSG module. Detailed discussion is provided in lines 330–334. Across different datasets, the optimal result is achieved when $r$ is around 100%, and this range has been verified in Figure 7. Our method involves only two key hyperparameters, $r$ and $\gamma$, both of which have been thoroughly analyzed. To compare with existing methods, we have counted the number of hyperparameters that critically affect the performance of each method:
>
> | Method | BETA [1] | SEAL [2] | RFC [3] | ADU [4] | AEM [5] | RrED |
> | ---- | ---- | ---- | ---- | ---- | ---- | ---- |
> | Key hyperparameters | 4 | 2 | 2 | 3 | 3 | 2 |
>
> According to the statistics of key parameters, our method requires relatively fewer key hyperparameters, which makes it more adaptable to practical scenarios without extensive hyperparameter tuning. Furthermore, as shown in Table 7, our method improves performance while reducing time consumption compared to prior works, thus offering a more cost-effective solution and demonstrating greater suitability for practical applications:
>
> | Method | Space(MiB) | Time(s/epoch) | Accuracy(%) |
> | --- | --- | --- | --- |
> | DINE | 9881MiB | 124s | 75.6 |
> | BETA | 20247MiB | 1101s | 85.1 |
> | SEAL | Over 24G | - | 89.2 |
> | AEM | 13747MiB | 672s | 89.3 |
> | RrED (Stage 1) | 17654MiB | 312s | 89.4 |
> | RrED (Stage 2) | 11721MiB | 201s | 91.2 |
>
> Q4: Lack of quantitative and qualitative analysis of key innovative designs. After hard reading, it can find that the Feature Sensitive Generation, Finetuning diffusion models, and Sample-version Interaction are key innovative modules. Besides final classification performance, we don't know what intuitive and direct result these designs lead to. For example, what do the fused examples after Feature Sensitive Generation really look like? Although Figure 7 gives some samples generated by diffusion model, it is hard to know the difference before and after finetuning.
>
> A4: Please refer to Figure 3, Figure 7, and Table 6, We have conducted detailed quantitative and qualitative analyses on each innovation module. Figure 3 provides an intuitive visualization of the qualitative and quantitative impact of each module using Grad-CAM technique. Table 6 presents detailed quantitative ablation results, while Figure 7 illustrates the quality differences in images generated by the diffusion model before (Strength = 0.6) and after fine-tuning (FT). For the Feature Sensitive Generation (FSG) by reviewers, “w/o FSG” represents the impact on the model after the removal of FSG. In addition, considering the reviewer’s interest in the FSG, we plan to provide more detailed visualizations of images before and after FSG processing in the final version, to more clearly illustrate the contribution of this module.
>
> Q5: How to effectively manage so many modules to make it stable in actual scenarios?
>
> A5: Please refer to A2, we have described how to effectively manage our modules, while we have encapsulated the entire method into two functions, each corresponding to one learning stage, making it easier to deploy and adapt in actual scenarios. Please refer to A3, our method combines self-adaptation with minimal hyperparameter tuning, requiring only two hyperparameters to achieve stable performance in actual scenarios. Compared to previous BUDA methods, our method is relatively more transferable.
>
> Q6: Supplement analysis and intuitive examples for the immediate and innovative designs.
>
> A6: Please refer to A4, we sincerely thank the reviewer for the valuable feedback. To more clearly illustrate the contribution of this module, we plan to provide more detailed visualizations of images before and after FSG processing and more analysis and intuitive examples in the final version.
>
> We appreciate the reviewer’s feedback and hope our response has addressed the reviewer’s concerns.
>
> [1] Divide to adapt: Mitigating confirmation bias for domain adaptation of black-box predictors. In ICLR, 2023.
>
> [2] A Separation and Alignment Framework for Black-Box Domain Adaptation. In AAAI, 2024.
>
> [3] Reviewing the forgotten classes for domain adaptation of black-box predictors. In AAAI, 2024.
>
> [4] ADU: Adaptive Detection of Unknown Categories in Black-Box Domain Adaptation. In CVPR, 2025.
>
> [5] Adversarial Experts Model for Black-box Domain Adaptation. In ACMMM, 2024.

---

> > ### Comment · Reviewer_mM7d · 2025-08-03
> >
> > Thanks for the active response and hard work from authors. Some of my concerns have been addressed, such as the whole system pipeline and modular design. However, there are still several points remained.
> >
> > As for the problem of sensitive hyperparameter tuning due to complicated design:
> > 1. You claimed that Figure 7 compares the two datasets, VisDA-17 and Office-Home (A→C). But actually it is not. The figure just displays some examples generated by the diffusion model. Also, I don't find any other figure matching your description.
> > 2. Even if there is one, I think it is not solid that the absolute number of samples for different datasets is the main factor. If it is, a further analysis about the influence of data amount or distribution on the hyperparameter selections is still needed.
> >
> > As for the problem of analysis for innovative modules, I understand that now it is not allowed to add visual examples in discussion. But there are still some concerns, especially for the Grad-CAM results in Figure 3:
> > 1. In general, Grad-CAM is a specific method to compute and visualize the gradient or back-propagation reliance of a certain block to final prediction. But in the paper, I don't notice any direct or indirect connection between your method and Grad-CAM in methdology.
> > 2. Grad-CAM is a sample-based method rather than a group-based or class-based. It means that different samples might have extremely different results from the same model and same class. In other words, if you pick a sample for which your method RrED misclassified but a variant ablation classified correctly, you can get a totally opposite conclusion.

---

> > > ### Author Response · Authors · 2025-08-04
> > > **Further rebuttal**
> > >
> > > Q7: You claimed that Figure 7 compares the two datasets ... any other figure matching your description.
> > >
> > > A7: Due to our oversight, Figure 7 was incorrectly referenced. In fact, Figure 4 should be instead of Figure 7 in the previous A3. We hereby issue the following correction to A3 as:
> > >
> > > “In Figure 7, we compare VisDA-17 and Office-Home (A→C) ... this range has been verified in Figure 7” **should be corrected to** “In Figure 4, we compare VisDA-17 and Office-Home (A→C), where the number of target domain samples differs significantly: VisDA-17 contains around 55K samples, while Office-Home has only about 4K. As such, a large performance gap between the two is expected. In addition, the hyperparameter $r$, which determines the ratio between regions of interest and non-interest, plays a critical role in the effectiveness of our FSG module. Detailed discussion is provided in lines 330–334. Across different datasets, the optimal result is achieved when $r$ is around 100%, and this range has been verified in Figure 4”.
> > >
> > > As shown in Figure 4, we present the performance under varying values of $\gamma$ and $r$ to evaluate the sensitivity of these hyperparameters. As we have described, the detailed discussion of $r$ is provided in lines 330–334 and the hyperparameter sensitivity visualization is shown in Figure 4, which has been verified on two datasets.
> > >
> > > We sincerely apologize for this oversight and are sorry for misleading the reviewer mM7d.
> > >
> > > Q8: Even if there is one, I think it is ... the hyperparameter selections is still needed.
> > >
> > > A8: Since the parameter $r$ determines the ratio between regions of interest and non-interest, which directly affects the effectiveness of the FSG module, we conducted a data amount ablation study to address the reviewer’s concern. Specifically, on the VisDA-17 dataset, we followed [1] for few-shot domain adaptation, where only 1% of the target samples were used for adaptation training, and the remaining samples were used for testing. The results of data amount ablation are as follows:
> > >
> > > | $r$=0 | =40% | =80% | =120% | =160% | =200% |
> > > | --- | --- | --- | --- | --- | --- |
> > > | 80.5 | 81.9 | 82.4 | 82.6 | 80.7 | 77.9 |
> > >
> > > The experimental results demonstrate that data amount has a significant impact on the hyperparameter $r$: when the training data is reduced, the model becomes less sensitive (Only the maximum 4.7% performance difference) to the choice of $r$, but its overall performance also degrades noticeably. Therefore, the absolute number of samples in datasets significantly influences the sensitivity of $r$, which supports our previous statement.
> > >
> > > Q9: In general, Grad-CAM is a ... your method and Grad-CAM in methdology.
> > >
> > > A9: As shown in lines 307-309, Grad-CAM highlights that each component of our algorithm plays a vital and irreplaceable role. Our method is directly related to Grad-CAM in methodology. We provide more detailed descriptions from both the background and functional perspectives.
> > >
> > > For the background, RrED is inspired by human decision-making systems and aims to refine model reasoning through a two-stage correction process. In computer vision, the model’s reasoning process mirrors human decision-making by utilizing existing knowledge and feature similarity computations to determine whether targets in complex scenes meet the task requirements. As indicated in the caption of Figure 2, RrED’s two-stage correction focuses on improving the diffusion model’s reasoning ability, thereby guiding and enhancing the reasoning of the target model. Therefore, verifying whether the model’s reasoning ability improves during optimization is a central focus of our experiments. To verify the model's reasoning ability, we introduce Grad-CAM, which is well-known for validating the reasoning ability of models.
> > >
> > > For the functional, we employ Grad-CAM in Figure 3 to highlight the vital and irreplaceable roles plays in the overall performance, with detailed explanations provided in lines 318-323. As shown in Figure 3, Grad-CAM clearly demonstrates that our model exhibits stronger reasoning capabilities, better object recognition, and more precise capture of fine-grained features in target samples compared with existing SOTA methods.
> > >
> > > Q10: Grad-CAM is a sample-based ... you can get a totally opposite conclusion.
> > >
> > > A10: The true class labels of each target image are annotated on the far left of Figure 3. In Figure 3, after training, our method significantly improves object recognition and captures fine-grained features of the targets more accurately. Furthermore, we conducted supplementary experiments to validate the prediction outputs, which were found to be fully consistent with the true labels on the far left. This demonstrates that our method’s assumptions align fully with the conclusion.
> > >
> > > Sincerely, we hope our response has addressed all concerns and wins the positive score of the reviewer.
> > >
> > > [1] Confidence-based visual dispersal for few-shot unsupervised domain adaptation. In ICCV, 2023.

---

> > > > ### Comment · Reviewer_mM7d · 2025-08-04
> > > >
> > > > Thanks for the further response from authors. For the concerns about influence of dataset scale on hyperparameter selection, although it is not perfect, at least the trend is consistent within the same dataset and across different datasets.
> > > >
> > > > As for the concerns of Grad-CAM, I don't think there is already a consensus for the reasoning definition or pipeline in vision area. Classification is not the full picture of vision tasks, either. Segmentation, vision-language alignment .etc still have different view points.Further research and justification are still needed.
> > > >
> > > > Anyway, in view of all factors above and from other reviewers, I decided to improve the rating to borderline accept.

---

> ### Author Response · Authors · 2025-08-05
> **Thank you for your active involvement and prompt feedback**
>
> Thank you for your active involvement and prompt feedback during the discussion phase. We are pleased to know that our work has received your acceptance.
>
> For the reasoning definition of Grad-CAM, we provide the following explanation to address the reviewer’s concern: in human decision-making systems, the definition of reasoning ability refers to the capacity of individuals to make further judgments about target objects by leveraging prior knowledge and logical analysis based on partial observations when confronted with complex information and dynamic environments. Similarly, in computer vision, the model’s reasoning process mirrors human decision-making by utilizing existing knowledge and feature similarity computations to determine whether targets in complex scenes meet the task requirements. This perspective aligns with the explanations provided by Grad-CAM, where models reason about predicted image classes based on convolutional feature maps, analogous to how humans reason about image categories through attention maps. The reasoning definition is based on the well-known Grad-CAM technique in computer vision, and our work further integrates the observations of the human decision-making system to refine this definition and enhance the model’s reasoning capabilities.
>
> For “classification is not the full picture of vision tasks, either. Segmentation, vision-language alignment, etc still have different view points. Further research and justification are still needed”, our theoretical justifications have demonstrated the theoretical feasibility of our method RrED for the BUDA task. To further validate its applicability across different tasks, we plan to extend RrED to more practical segmentation and detection tasks in future work. Please look forward to our upcoming research.
>
> Your constructive review has undoubtedly enhanced the quality of our paper, and we sincerely appreciate your contribution.

---

### Official Review · Reviewer_PJJd · 2025-06-25

**Clarity:** 3
**Significance:** 3
**Originality:** 3
**Rating:** 5
**Confidence:** 4

**Summary:**

This work introduces a novel diffusion-based method called Reifying-reasoning Errors of Diffusion (RrED) for Black-box Unsupervised Domain Adaptation (BUDA). RrED enhances target models by decomposing text and visual encoders, using a feature-sensitive module for visual enhancement and multi-modal joint fine-tuning. It prioritizes BUDA tasks and leverages differential reasoning to rectify errors, showing improved performance in reasoning and generalization across four benchmark datasets. The proposed method is relatively cumbersome and complex, but has achieved good performance.

**Questions:**

1) Please expain what is the mean of lines 177-182.  Meanwhile, it is evident that the classifier p of the diffusion model heavily relies on the pre-trained model on which dataset, therefore it is necessary to introduce the pre-trained diffusion model.

2) Please explain Eq (6). Why can category related areas be selected.

3) From my understanding, the first step is essentially using the diffusion model to update the multi model and local model. So, how can we ensure that the diffusion model is correct in this area, as the target domain samples may not have participated in training the diffusion model.

4) This work is designed with many loss functions, and the number of parameters discussed is far less than the actual number of loss functions. Although the authors fixed some parameters to 1, they still need to be discussed.

5) Please deeply explain Eq.(13). It is obvious that these two optimization items are conflicting.

**Ethical Concerns:**

["NO or VERY MINOR ethics concerns only"]

**Final Justification:**

I am generally satisfied with the responses, so I plan to maintain the previous score.

**Limitations:**

Yes

**Paper Formatting Concerns:**

It is still recommended to move the contribution to the main text and place the introduction of the diffusion model in the appendix. Also, what is Lemma 1 in Appendix doing? I'm not quite sure,  a more logical organization for this part is needed.

**Quality:**

3

**Strengths And Weaknesses:**

Strengths:
1)  The proposed FSG block evaluates feature-sensitive regions by comparing global and local image regions, enhancing generalization without losing discriminative ability.

2) The first to use diffusion model for black box domain adaptation problem.  SOTA performance.

Weaknesses:
1) Writing needs further improvement. For example, Figure 1 is too complex to express the core idea. Figure 2 is also too complex. At the same time, the diffusion model should not be introduced too much, and the contribution should be written in the main text.

2) Introducing the diffusion model to assist in black box adaptation also brings in more knowledge. So in essence, there are unfairness in comparing with existing methods.

---

> ### Author Rebuttal · Authors · 2025-07-30
>
> We sincerely appreciate the reviewer’s constructive suggestions and have answered them as follows with our highest respect.
>
> Q1: Writing needs further improvement. For example, Figure 1 is too complex to express the core idea. Figure 2 is also too complex. At the same time, the diffusion model should not be introduced too much, and the contribution should be written in the main text.
>
> A1: We sincerely thank the reviewer for the valuable feedback and will continue to improve our writing. To improve the clarity and structure of the paper, we have simplified Figures 1 and 2, moved the contribution summary into the main text, and relocated the diffusion process to the Appendix.
>
> Q2: Introducing the diffusion model to assist in black box adaptation also brings in more knowledge. So in essence, there are unfairness in comparing with existing methods.
>
> A2: Our RrED is the first work that applies the diffusion model to high-security BUDA tasks innovatively. Since no prior work shares exactly the same experimental configuration, a direct and perfectly fair comparison is not achievable. Accordingly, we perform comparisons with representative methods under both high- and low-challenge settings to indirectly demonstrate the fairness of our evaluation. Please refer to Table 1, compared to Diffusion-based DA, which has access to both source and target domains, our RrED offers stronger privacy protection by operating without any access to source knowledge. Compared to Black-box DA, which does not utilize diffusion models, RrED leverages knowledge from diffusion models to achieve significant performance improvements. Whether compared with the less challenging Diffusion-based DA or the more challenging Black-box DA, RrED has achieved significant improvements. In particular, compared to the less challenging Diffusion-based DA method DAD, RrED still achieved a maximum improvement of 13.1% on Office-Home without being able to obtain source knowledge. Therefore, our experiment achieved a fair comparison.
>
> Q3: Please explain what is the mean of lines 177-182. Meanwhile, it is evident that the classifier p of the diffusion model heavily relies on the pre-trained model on which dataset, therefore it is necessary to introduce the pre-trained diffusion model.
>
> A3: We provide detailed explanations for lines 177-182 from the perspectives of motivation, process, and result.
>
> For the motivation of lines 177-182, as shown in Figure 7, when the noise level is too low, the images generated by the diffusion model are too similar to the target domain images, providing limited benefit for enhancing the model's reasoning ability. When the noise level is too high, the images generated by the diffusion model differ drastically from those in the target domain and may even contain unrelated objects. This leads us to a key question: how can we effectively utilize the diffusion model to guide the target model in enhancing its reasoning ability while preventing its potential negative effects? These motivations are described in lines 693-699.
>
> For the process of lines 177-182, in our analysis of the Stable Diffusion model, we observed that it employs a CLIP-based enhanced text encoder to interpret textual inputs. Building upon this, we extracted the image encoder from CLIP and combined it with the enhanced text encoder from Stable Diffusion to construct a new text-image model. Inspired by recent advances in prompt learning (PL), as shown in Eq. (8), we insert learnable continuous vectors before the text encoder. During the first training stage, only these vectors are updated, while all other parameters in the new text-image model remain frozen. Since the learnable vectors are not yet well-optimized in the first stage and the Stable Diffusion model is not frequently invoked, we delay integrating the modified text encoder. Then, we replace the original text encoder in the Stable Diffusion model with the one incorporating the learnable vectors at the beginning of the second stage.
>
> For the result of lines 177-182, as shown in Figure 7 and Table 7, this design significantly reduces computational overhead during training while enabling the diffusion model to better comprehend textual inputs, thereby generating higher-quality images and ultimately achieving substantial performance improvements.
>
> For “it is evident that the classifier p of the diffusion model heavily relies on the pre-trained model on which dataset, therefore it is necessary to introduce the pre-trained diffusion model”, as shown in Figure 6, the classifier $p$ within the diffusion model contains richer semantic information compared to the black-box predictor. However, directly fine-tuning the diffusion model’s parameters is computationally expensive for our BUDA downstream tasks. Therefore, we uniformly adopt the default Stable Diffusion model as the diffusion module without additional pretraining, and employ a fixed diffusion classifier only during the first stage to provide guidance. As shown in Figure 8, by the end of the first stage, the target model already outperforms the diffusion classifier, making further use of it unnecessary. We then proceed to the second stage, where the target model is optimized independently. Our design achieves remarkable performance without any pretraining of the diffusion model on either the target or source domain samples. Due to the high computational cost of dataset-specific pretraining, we are currently unable to perform such pretraining for the Stable Diffusion model. We plan to further explore this direction once sufficient computational resources become available.
>
> Q4: Please explain Eq (6). Why can category related areas be selected.
>
> A4: In Eq. (6), FSG evaluates the feature-sensitive regions of the model by evaluating the global image and comparing it with each local region, leveraging the local regions and their adjacent contextual information. The category related areas are determined by the target model during training, and this selection process is obtained through evaluation calculation. As shown in Figure 3, the selection process is dynamic. The model evaluates local regions to identify feature-sensitive areas, and then leverages the features from these regions to determine class-relevant information.
>
> Q5: From my understanding, the first step is essentially using the diffusion model to update the multi model and local model. So, how can we ensure that the diffusion model is correct in this area, as the target domain samples may not have participated in training the diffusion model.
>
> A5: Due to limited computational resources, we are unable to fully train the diffusion model on the target domain, and thus cannot ensure the accuracy of its classification outputs. As shown in Figure 6, the diffusion model achieves a classification accuracy of 56.3%, higher than the black-box predictor but still far from sufficient for reliable BUDA tasks. To address this limitation, we reduce the noise interference of the diffusion model from two directions while ensuring that the diffusion model correctly guides the learning of the target model: (1) Our framework is designed in two stages: In the first stage, we leverage the rich semantic knowledge embedded in a fixed diffusion model to guide the initial learning of the target model. In the second stage, we discard the diffusion model’s predictions and instead use high-quality images generated by the diffusion model with a fine-tuned text encoder to further train the target model. (2) We propose Feature-Sensitive Generation, which integrates diffusion outputs through image fusion to serve as a buffer mechanism. This helps the model focus on class-discriminative regions more effectively.
>
> Q6: This work is designed with many loss functions, and the number of parameters discussed is far less than the actual number of loss functions. Although the authors fixed some parameters to 1, they still need to be discussed.
>
> A6: We sincerely thank the reviewer for the valuable feedback. As shown in lines 594-595, “Following previous BUDA works, as presented in Eq. (2), $\lambda$ is set to 1 in the BUDA task”, Eq. (2) is a task-specific loss. To verify its effect on our method, we supplemented the following experiments on the VisDA-17 dataset:
>
> | $\lambda$=0.2 | $\lambda$=0.5 | $\lambda$=1 | $\lambda$=1.5 | $\lambda$=2 | $\lambda$=3 |
> | --- | --- | --- | --- | --- | --- |
> | 88.4 | 89.9 | 91.2 | 91.4 | 90.9 | 90.8 |
>
> Experimental results show that appropriately tuning the weight of the task-specific loss can improve model performance to some extent.
>
> Q7: Please deeply explain Eq.(13). It is obvious that these two optimization items are conflicting.
>
> A7: In Eq. (13), our proposed interactive optimization learning can be divided into two optimization items. Two items form a contrastive learning, the first term aims to correct model reasoning errors by aligning and clustering features based on the similarity between source and generated samples. Meanwhile, the second term enhances the feature discrepancies among samples to prevent overfitting caused by overly rapid alignment from the first term. The design principle of using one term for task execution and another to regulate its learning pace is common in contrastive learning, and our implementation in Eq. (2) presents an innovative extension of this principle.
>
> Q8:  Also, what is Lemma 1 in Appendix doing? I'm not quite sure, a more logical organization for this part is needed.
>
> A8: In our paper, we use Lemma 1 to demonstrate the rationality of the BUDA setting, clarify the common principles underlying BUDA methods, and explain why our approach and other BUDA methods are applicable in this context. Lemma 1 serves Eq. (18)-(21). As shown in lines 597-598, our proof fills the theoretical knowledge gap regarding the black-box predictor in previous BUDA works.
>
> We appreciate the reviewer’s feedback and hope our response has addressed the reviewer’s concerns.

---

### Official Review · Reviewer_Wd9F · 2025-07-06

**Clarity:** 3
**Significance:** 3
**Originality:** 3
**Rating:** 5
**Confidence:** 4

**Summary:**

This paper proposes RrED, a novel two-stage framework for black-box Unsupervised Domain Adaptation (BUDA), which, for the first time, integrates diffusion models as a semantic knowledge base to guide adaptation; the framework consists of Diffusion-Target model Rectification (DTR), which aligns the target model with diffusion-generated semantic features, and Self-Rectifying Reasoning Model (SRM), which enhances the target model's reasoning ability through self-corrective learning, collectively enabling effective reasoning error correction and achieving superior performance over existing state-of-the-art methods.

**Questions:**

See weaknesses

**Ethical Concerns:**

["NO or VERY MINOR ethics concerns only"]

**Final Justification:**

I read the replies and thanks for authors' hard work and detailed explanation, it's more clear now.

**Limitations:**

see weaknesses

**Quality:**

3

**Strengths And Weaknesses:**

Strength:
    1:RrED is the first to apply the diffusion model to the highly secure BUDA task and innovatively fine-tuned the text encoder of the diffusion model. This resolves the issue that existing diffusion-based domain adaptation methods mainly focus on image encoders while neglecting the nuances of text.
    2:RrED significantly outperforms previous state-of-the-art methods on all four benchmark datasets, demonstrating its effectiveness in enhancing model reasoning and generalization capabilities.


Weakness:
    1:It remains unclear how sensitive the proposed RrED framework is to the choice of diffusion backbone. For instance, would replacing the default diffusion model with alternatives such as Stable Diffusion or DALL·E yield comparable results, or introduce significant performance discrepancies? A systematic analysis of this dependency would strengthen the empirical foundation of the method and inform its generalizability.

    2:RrED is tailored for the BUDA setting, but it remains unclear whether its core components or two-stage framework can generalize to other domain adaptation scenarios such as open-set, or partial adaptation. Exploring its applicability beyond BUDA would further demonstrate the method's versatility.

    3：The paper lacks visualizations or quantitative analyses that assess the prediction quality of the black-box predictor.   In particular, it does not provide dedicated figures or experiments illustrating how varying levels of quality—such as low accuracy, biased predictions, or noisy outputs—affect the downstream performance of RrED.

---

> ### Author Rebuttal · Authors · 2025-07-30
>
> We sincerely appreciate the reviewer’s constructive suggestions and have answered them as follows with our highest respect.
>
> Q1: It remains unclear how sensitive the proposed RrED framework is to the choice of diffusion backbone. For instance, would replacing the default diffusion model with alternatives such as Stable Diffusion or DALL·E yield comparable results, or introduce significant performance discrepancies? A systematic analysis of this dependency would strengthen the empirical foundation of the method and inform its generalizability.
>
> A1: In our experiments, we have analyzed backbone sensitivity by conducting both qualitative and quantitative evaluations, where the default Stable Diffusion model was used in place of our fine-tuned version. In Figure 3, “w/o Fine-tuning” refers to using the default text encoder of the Stable Diffusion model without any fine-tuning, while “w/o FSG” indicates directly using the images generated by the default Stable Diffusion model without any further processing. As shown in Figure 3, we have performed qualitative and quantitative ablation evaluations on the VisDA-17 dataset to determine the sensitivity of each module to model improvements. The visualization also demonstrates that the model captures target features more accurately after improvements. Furthermore, based on Table 6, we supplement the results of using only the default Stable Diffusion backbone (w/o Fine-tuning and FSG) to further verify the system analysis of dependencies:
>
> | Office-31 | A→D | A→W | D→A | D→W | W→A | W→D | Mean | VisDA-17 (Mean) |
> | ---- | ---- | ---- | ---- | ---- | ---- | ---- | ---- | ---- |
> | RrED w/o Fine-tuning | 96.8 | 94.1 | 82.5 | 99.1 | 84.1 | 99.8 | 92.7 | 88.7 |
> | RrED w/o FSG | 73.1 | 85.7 | 69.7 | 95.3 | 65.2 | 97.9 | 81.2 | 76.8 |
> | RrED w/o Fine-tuning and FSG | 71.7 | 84.9 | 70.2 | 94.6 | 64.4 | 98.2 | 80.6 | 75.9 |
> | RrED | 97.8 | 95.9 | 83.7 | 99.1 | 84.5 | 99.8 | 93.5 | 91.2 |
>
> In the above table, we conducted a comparison in two different datasets Office-31 and VisDA-17. The results indicate that directly applying the default Stable Diffusion model, without adaptation to the downstream task, leads to a sharp performance drop. In contrast, our method exhibits highly task-aware sensitivity to the structural characteristics of the Stable Diffusion model, enabling it to better leverage its semantic knowledge for downstream BUDA tasks. We plan to include these analyses and experiments in the final version.
>
> Q2: RrED is tailored for the BUDA setting, but it remains unclear whether its core components or two-stage framework can generalize to other domain adaptation scenarios such as open-set, or partial adaptation. Exploring its applicability beyond BUDA would further demonstrate the method's versatility.
>
> A2: To explore the applicability of RrED in other tasks and further demonstrate its versatility, we conducted experiments in open-set black-box adaptation scenarios. We integrate our RrED into [1] to enable domain adaptation under the open-set black-box adaptation setting on Office-Home, and the results are as follows:
>
> | Office-Home | A→C | A→P | A→R | C→A | C→P | C→R | P→A | P→C | P→R | R→A | R→C | R→P | Mean |
> | ---- | ---- | ---- | ---- | ---- | ---- | ---- | ---- | ---- | ---- | ---- | ---- | ---- | ---- |
> | UB2DA | 60.9 | 69.6 | 76.3 | 74.4 | 69.2 | 76.5 | 74.5 | 60.3 | 76.2 | 74.1 | 62.0 | 71.1 | 70.4 |
> | ADU | 61.2 | 72.7 | 77.9 | 70.3 | 72.5 | 77.3 | 75.9 | 62.0 | 84.7 | 73.2 | 64.1 | 74.9 | 72.2 |
> | RrED | 69.1 | 81.6 | 84.3 | 77.9 | 78.2 | 83.1 | 81.8 | 67.3 | 87.6 | 79.6 | 70.9 | 80.4 | 78.5 |
>
> As shown in the table above, even under a different setting, RrED continues to demonstrate strong performance and significantly outperforms open-set black-box adaptation SOTA methods [1, 2].
>
> Q3: The paper lacks visualizations or quantitative analyses that assess the prediction quality of the black-box predictor. In particular, it does not provide dedicated figures or experiments illustrating how varying levels of quality—such as low accuracy, biased predictions, or noisy outputs—affect the downstream performance of RrED.
>
> A3: In our paper, we have demonstrated through extensive quantitative and qualitative experiments that the black-box predictor has inherent limitations in prediction quality. In Tables 2-4 and Figure 3, "Source-only" refers to using the black-box predictor to evaluate the predicted target samples. In addition, we visualize the features of the black-box predictor using t-SNE in Figure 8. To enhance understanding, we provide the following additional explanations: As shown in Figure 8, the black-box predictor fails to effectively separate and cluster target sample features, with samples from different classes heavily entangled. This confusion introduces noisy signals during target model training, thereby hindering effective adaptation. We plan to include these explanations in the final version.
>
> We appreciate the reviewer’s feedback and hope our response has addressed the reviewer’s concerns.
>
> [1] On universal black-box domain adaptation. arXiv, 2021.
>
> [2] ADU: Adaptive Detection of Unknown Categories in Black-Box Domain Adaptation. In CVPR, 2025.

---

> > ### Comment · Reviewer_Wd9F · 2025-08-06
> >
> > Thanks for the detailed explanation by the authors. It's much clearer now. So I changed my rating to reflect it.

---

### Comment · Area_Chair_vnyW · 2025-08-03
**Reminder: Discussion and Final Rating Update**

Dear Reviewers,

As we are now midway through the discussion phase, I would like to kindly remind you to review the authors' rebuttal and participate in the discussion. Please also update your review with a final rating accordingly.

Thank you very much for your time and valuable contributions to the review process.

Best regards,

Area Chair

---

### Decision · Program_Chairs · 2025-09-17

**Decision:**

Accept (poster)

**Comment:**

This paper has received consistent positive ratings of 4, 4, 5, and 5. The reviewers raised concerns regarding backbone sensitivity, the specificity of the proposed design, fairness in comparisons, and issues with writing quality. The authors have addressed all of these points in their rebuttal, and the reviewers expressed satisfaction with the responses. While the writing issues can certainly be improved in the revised version, they currently hinder readability and limit the potential for a higher score. Given the consistent positive evaluations and the reviewers’ satisfaction with the rebuttal, I see no compelling reason to overturn their acceptance recommendations. Therefore, I recommend acceptance.